# Comparative analysis of the complete chloroplast genomes of thirteen *Bougainvillea* cultivars from South China with implications for their genome structures and phylogenetic relationships

**Xiao-Ye Wu[1], He-Fa Wang[2], Shui-Ping Zou[1], Lan Wang[1], Gen-Fa Zhu[3]\*, Dong-Mei Li [3]\***

**1** Research Institute of Living Environment, Guangdong Bailin Ecology and Technology Co., Ltd., Dongguan, China, **2** Xiamen Qianrihong Horticulture Co., Ltd., Xiamen, China, **3** Guangdong Key Lab of Ornamental Plant Germplasm Innovation and Utilization, Environmental Horticulture Research Institute, Guangdong Academy of Agricultural Sciences, Guangzhou, China

\* genfazhu@163.com (G-FZ); biology.li2008@163.com (D-ML)

**Data Availability Statement:** Data Availability Statement: The data presented in this study were

## Abstract

*Bougainvillea* spp., belonging to the Nyctaginaceae family, have high economic and horticultural value in South China. Despite the high similarity in terms of leaf appearance and hybridization among *Bougainvillea* species, especially *Bougainvillea × buttiana*, their phylogenetic relationships are very complicated and controversial. In this study, we sequenced, assembled and analyzed thirteen complete chloroplast genomes of *Bougainvillea* cultivars from South China, including ten *B. × buttiana* cultivars and three other *Bougainvillea* cultivars, and identified their phylogenetic relationships within the *Bougainvillea* genus and other species of the Nyctaginaceae family for the first time. These 13 chloroplast genomes had typical quadripartite structures, comprising a large single-copy (LSC) region (85,169–85,695 bp), a small single-copy (SSC) region (18,050–21,789 bp), and a pair of inverted-repeat (IR) regions (25,377–25,426 bp). These genomes each contained 112 different genes, including 79 protein-coding genes, 29 tRNAs and 4 rRNAs. The gene content, codon usage, simple sequence repeats (SSRs), and long repeats were essentially conserved among these 13 genomes. Single-nucleotide polymorphisms (SNPs) and insertions/deletions (indels) were detected among these 13 genomes. Four divergent regions, namely, *trnH-GUG_psbA*, *trnS-GCU_trnG-UCC-exon1*, *trnS-GGA_rps4*, and *ccsA_ndhD*, were identified from the comparative analysis of 16 *Bougainvillea* cultivar genomes. Among the 46 chloroplast genomes of the Nyctaginaceae family, nine genes, namely, *rps12*, *rbcL*, *ndhF*, *rpoB*, *rpoC2*, *ndhI*, *psbT*, *ycf2*, and *ycf3*, were found to be under positive selection at the amino acid site level. Phylogenetic relationships within the *Bougainvillea* genus and other species of the Nyctaginaceae family based on complete chloroplast genomes and protein-coding genes revealed that the *Bougainvillea* genus was a sister to the *Belemia* genus with strong support and that 35 *Bougainvillea* individuals were divided into 4 strongly supported clades, namely, Clades I, II, III and IV. Clade I included 6 individuals, which contained 2 cultivars, namely, *B. × buttiana* 'Gautama's Red' and *B. spectabilis* 'Flame'. Clades II only

submitted to the NCBI repository (https://www.ncbi.nlm.nih.gov) under accession numbers OR344366–OR344378.

**Funding:** This research was financially supported by the Collection, Identification and Utilization of New and Superior Flower Germplasm Resources (2023-2025), Science and Technology Program from Forestry Administration of Guangdong Province (2024KJQT0014) and the Guangdong Province Modern Agriculture Industry Technical System–Flower Innovation Team Construction Project (2023KJ121). The collection, identification and utilization of new and superior flower germplasm resources (2023-2025) and Science and Technology Program from Forestry Administration of Guangdong Province (2024KJQT0014) were funded by Guangdong Bailin Ecology and Technology Co., Ltd. The Guangdong Province Modern Agriculture Industry Technical System–Flower Innovation Team Construction Project (2023KJ121) was funded by the Environmental Horticulture Research Institute, Guangdong Academy of Agricultural Sciences. There was no additional external funding received for this study.

**Competing interests:** The authors have declared that no competing interests exist.

contained *Bougainvillea spinosa*. Clade III comprised 7 individuals of wild species. Clade IV included 21 individuals and contained 11 cultivars, namely, *B. × buttiana* 'Mahara', *B. × buttiana* 'California Gold', *B. × buttiana* 'Double Salmon', *B. × buttiana* 'Double Yellow', *B. × buttiana* 'Los Banos Beauty', *B. × buttiana* 'Big Chitra', *B. × buttiana* 'San Diego Red', *B. × buttiana* 'Barbara Karst', *B. glabra* 'White Stripe', *B. spectabilis* 'Splendens' and *B. × buttiana* 'Miss Manila' sp. 1. In conclusion, this study not only provided valuable genome resources but also helped to identify *Bougainvillea* cultivars and understand the chloroplast genome evolution of the Nyctaginaceae family.

## Introduction

The Nyctaginaceae family, also called the four o'clock family, contains approximately 31 genera [1,2]. San Jiao Mei and Le Du Juan, which are well known in China, belong to the *Bougainvillea* genus of the Nyctaginaceae family. *Bougainvillea* plants are tropical and subtropical shrubs or small trees armed with simple or forked thorns, commonly with colorful bracts [2,3]. The colorful bracts surrounding small tubular flowers are often mistakenly treated as flowers. Due to the demand of the commercial market, garden growers have obtained new cultivars with bright bracts through hybridization or grafting [4,5].

To date, more than 200 *Bougainvillea* cultivars have been produced and introduced to China. *Bougainvillea* cultivars with large bracts of various colors have been seen in many cities of South China, such as Zhangzhou of Fujian, Guangzhou of Guangdong, Haikou of Hainan, and Nanning of Guangxi. *Bougainvillea × buttiana* cultivars have also been used in many cities in South China. *B. × buttiana* is named a new species based on a plant cultivated in the Singapore Botanical Garden [4]. It was originally from a garden in Cartagena, Colombia, and was introduced to Trinidad in 1910 as the cultivar 'Mrs. Butt'. It is presumed by Gillis [5] to be a hybrid between *Bougainvillea peruviana* and *B. glabra*. These *Bougainvillea* cultivars have been widely used for horticultural landscaping in cities of South China. However, identification of these *Bougainvillea* cultivars based mainly on leaf morphology has been challenging because of the high similarity of their leaf appearances [2,3].

In previous studies, although the phylogenetic relationships of the Nyctaginaceae family, including the *Bougainvillea* genus, were identified using several chloroplast genes (*ndhF*, *rps16*, and *rpl16*) and one nuclear region (*ITS*), low-resolution branches among different genera existed [1,6]. With recent advancements in sequencing, complete chloroplast genome sequencing has become convenient. Complete chloroplast genomes have been extensively used for phylogenetic analyses of ornamental plants, such as Caryophyllales [7], *Aglaonema* [8] and *Hyacinthus* [9]. More recently, the phylogenetic relationships of wild *Bougainvillea* species have been explored using complete chloroplast genomes [2,3,10–13] and even up to one hybrid cultivar [3]. However, the phylogenetic relationships of *B. × buttiana* cultivars and the molecular evolution of chloroplast genomes from the Nyctaginaceae family remain to be elucidated [2–5,9–13]. Therefore, it is worthwhile to investigate the phylogenetic relationships of *B. × buttiana* cultivars and the molecular evolution of chloroplast genomes in the Nyctaginaceae family.

In this study, the complete chloroplast genomes of thirteen *Bougainvillea* cultivars were newly sequenced, assembled and annotated. These thirteen cultivars from South China [14,15], included ten *B. × buttiana* cultivars, namely, *B. × buttiana* 'Mahara', *B. × buttiana* 'Gautama's Red', *B. × buttiana* 'California Gold', *B. × buttiana* 'Double Salmon', *B. × buttiana* 'Double Yellow', *B. × buttiana* 'Big Chitra', *B. × buttiana* 'Los Banos Beauty', *B. × buttiana*

'Barbara Karst', *B.* × *buttiana* 'San Diego Red', and *B.* × *buttiana* 'Miss Manila' sp. 1, which was one bud mutation armed with simple or no thorns and derived from *B.* × *buttiana* 'Miss Manila'; and three commonly used cultivars, namely, *B. glabra* 'White Stripe', *B. spectabilis* 'Flame' and *B. spectabilis* 'Splendens' (Fig 1). Then, we performed comparative genomics and phylogenomic analyses by integrating three published complete chloroplast genomes of *Bougainvillea* cultivars from the NCBI. In this study, five objectives were targeted: (1) to characterize and investigate the 13 newly sequenced complete chloroplast genome structures; (2) to detect variations in simple sequence repeats (SSRs), long repeats, and codon usage among these 13 chloroplast genomes; (3) to identify highly variable regions for potential DNA marker development among *Bougainvillea* cultivars; (4) to understand the molecular evolution of chloroplast genomes in the Nyctaginaceae family; and (5) to infer the phylogenetic relationships among *Bougainvillea* species and cultivars and other species of the Nyctaginaceae family.

## Materials and methods

### Plant materials, DNA extraction, and sequencing

Fresh leaves of twelve *Bougainvillea* cultivars, namely, *B.* × *buttiana* 'Mahara', *B.* × *buttiana* 'Gautama's Red', *B.* × *buttiana* 'California Gold', *B.* × *buttiana* 'Double Salmon', *B.* × *buttiana* 'Double Yellow', *B.* × *buttiana* 'Big Chitra', *B.* × *buttiana* 'Los Banos Beauty', *B. glabra* 'White Stripe', *B. spectabilis* 'Flame', *B. spectabilis* 'Splendens', *B.* × *buttiana* 'Barbara Karst', and *B.* × *buttiana* 'San Diego Red' (Fig 1, S1 Table), were collected from the Provincial Flower Germplasm Resources Bank of San Jiao Mei in Zhangzhou (117°37′47″E, 24°28′35″N), Fujian Province, China. One bud mutation armed with simple or no thorns and derived from *B.* × *buttiana* 'Miss Manila', given name, *B.* × *buttiana* 'Miss Manila' sp. 1 (Fig 1, S1 Table), was collected from the cultivation factoty of Zhangzhou (117°49′9″E, 24°31′33″N) in Xiamen Qianrihong Horticulture Co., Ltd, Fujian Province, China. Fresh leaves were quickly frozen on dry ice, sent to the laboratory of the Environmental Horticulture Research Institute (113°20′39″E, 23°8′51″N) at the Guangdong Academy of Agricultural Sciences, Guangzhou, China, and stored at −80° until use. Genomic chloroplast DNA was extracted from each sample using the modified sucrose gradient centrifugation method [16]. Then, the DNA quality and quantity were checked through agarose gel electrophoresis and the NanoDrop microspectrometer method, respectively. Each qualified DNA sample was sheared to fragments of approximately 350 bp. Short-insert (350 bp) paired-end libraries were constructed, and sequencing was performed on an Illumina NovaSeq 6000 platform with a paired read length of 150 bp (Biozeron, Shanghai, China). The raw data from each sample were checked using FastQC v. 0.11.9 (http://www.bioinformatics.babraham.ac.uk/projects/fastqc/), and adaptors and low-quality reads were subsequently deleted by Trimmomatic v. 0.39 [17] with default parameters. The remaining materials, including the leaves and DNA, were deposited in the laboratory of the Environmental Horticulture Research Institute (store sheet code: B2023, 113°20′39″E, 23°8′51″N), Guangdong Academy of Agricultural Sciences, Guangzhou, China, as vouchers (S1 Table).

### Chloroplast genome assembly and annotation

At least 5.6 Gb of clean data were obtained from each sample (S1 Table). Chloroplast genome assembly and annotation were conducted using previously reported methods [18]. In brief, the clean paired-end reads were assembled using GetOrganelle v. 1.7.6.1 [19] with default parameters. The published complete chloroplast genomes of *Bougainvillea peruviana* (GenBank MT407463) and *B. glabra* (GenBank MN888961) were used as references for sequence correction by Geneious Prime 2022.10 [20]. Gene annotation was carried out using GeSeq [21] and the online Dual Organellar Genome Annotator (DOGMA) [22] with default parameters. The

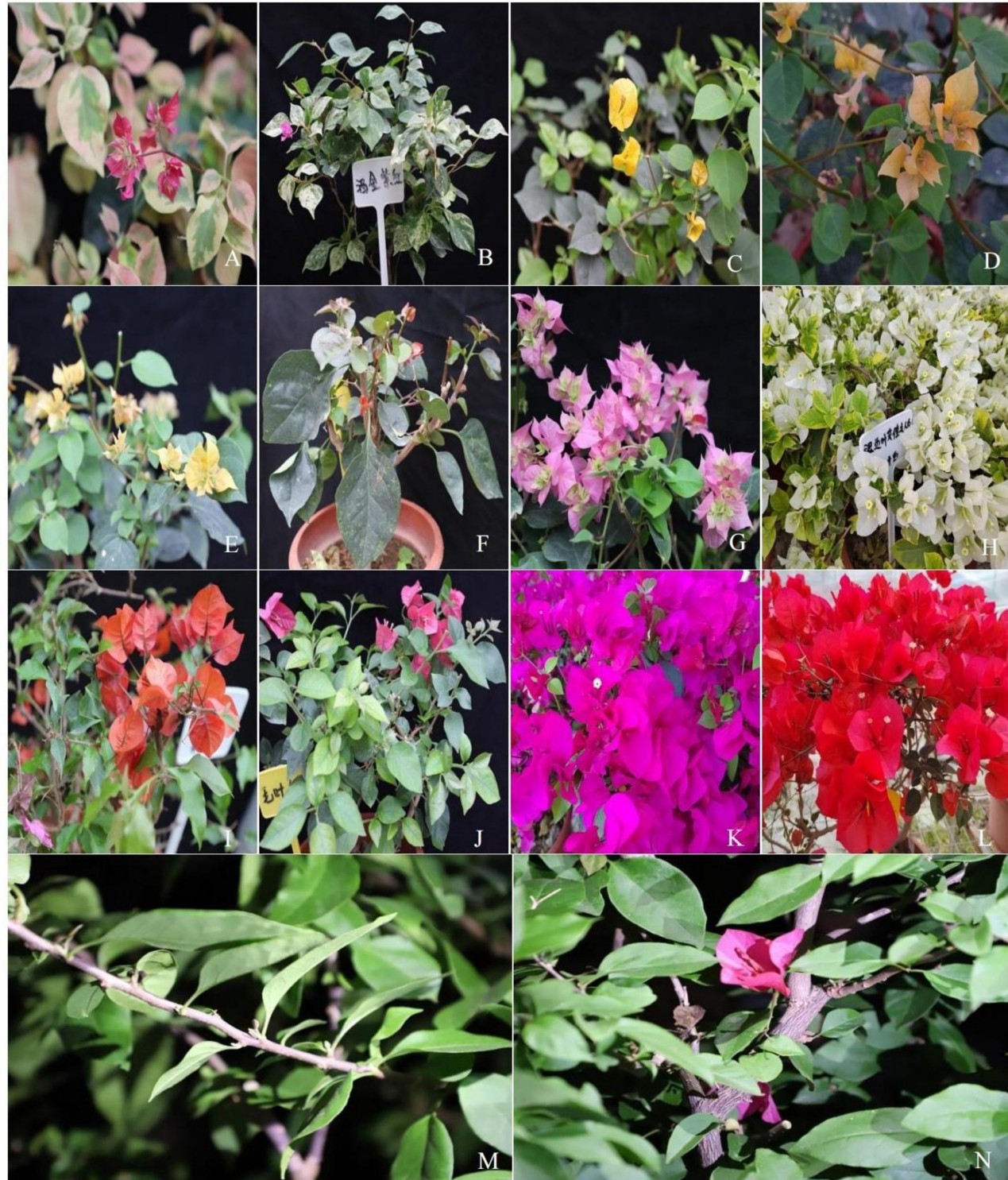

**Fig 1. Morphologies among 13 cultivars of the *Bougainvillea* genus.** A, *Bougainvillea×buttiana* 'Mahara'; B, *B.×buttiana* 'Gautama's Red'; C, *B. ×buttiana* 'California Gold'; D, *B.×buttiana* 'Double Salmon'; E, *B.×buttiana* 'Double Yellow'; F, *B. ×buttiana* 'Big Chitra'; G, *B.×buttiana* 'Los Banos Beauty'; H, *Bougainvillea glabra* 'White Stripe'; I, *Bougainvillea spectabilis* 'Flame'; J, *B. spectabilis* 'Splendens'; K, *B. × buttiana* 'Barbara Karst'; L, *B. × buttiana* 'San Diego Red'; M and N, *B. × buttiana* 'Miss Manila' sp. 1. armed with simple or no thorns, derived from *B. × buttiana* 'Miss Manila'.

**Table 1. Characteristics of the 13 newly sequenced complete chloroplast genomes of *Bougainvillea* cultivars.**

| Cultivars | GenBank accession | Size (bp) | LSC (bp) | SSC (bp) | IR (bp) | GC content (%) | | | | | Number of genes (different) | Number of CDSs (different) | Number of tRNAs (different) | Number of rRNAs (different) |
|---|---|---|---|---|---|---|---|---|---|---|---|---|---|---|
| | | | | | | Total | LSC | SSC | IR | CDS | | | | |
| *Bougainvillea × buttiana* 'Mahara' | OR344376 | 154,541 | 85,694 | 18,077 | 25,385 | 36.46 | 34.17 | 29.47 | 42.81 | 37.21 | 131 (112) | 86 (79) | 37 (29) | 8 (4) |
| *B. × buttiana* 'Gautama's Red' | OR344371 | 154,465 | 85,563 | 18,050 | 25,426 | 36.49 | 34.19 | 29.53 | 42.85 | 37.85 | 131 (112) | 86 (79) | 37 (29) | 8 (4) |
| *B. × buttiana* 'California Gold' | OR344368 | 154,542 | 85,695 | 18,077 | 25,385 | 36.46 | 34.17 | 29.47 | 42.81 | 37.19 | 131 (112) | 86 (79) | 37 (29) | 8 (4) |
| *B. × buttiana* 'Double Salmon' | OR344375 | 154,542 | 85,695 | 18,077 | 25,385 | 36.46 | 34.17 | 29.47 | 42.81 | 37.18 | 131 (112) | 86 (79) | 37 (29) | 8 (4) |
| *B. × buttiana* 'Double Yellow' | OR344373 | 154,542 | 85,695 | 18,077 | 25,385 | 36.46 | 34.17 | 29.47 | 42.81 | 37.18 | 131 (112) | 86 (79) | 37 (29) | 8 (4) |
| *B. × buttiana* 'Big Chitra' | OR344367 | 154,542 | 85,695 | 18,077 | 25,385 | 36.46 | 34.17 | 29.47 | 42.81 | 37.18 | 131 (112) | 86 (79) | 37 (29) | 8 (4) |
| *B. × buttiana* 'Los Banos Beauty' | OR344374 | 154,542 | 85,695 | 18,077 | 25,385 | 36.46 | 34.17 | 29.47 | 42.81 | 37.18 | 131 (112) | 86 (79) | 37 (29) | 8 (4) |
| *B. glabra* 'White Stripe' | OR344370 | 154,520 | 85,688 | 18,078 | 25,377 | 36.46 | 34.18 | 29.47 | 42.81 | 37.18 | 131 (112) | 86 (79) | 37 (29) | 8 (4) |
| *B. spectabilis* 'Flame' | OR344366 | 153,994 | 85,169 | 18,043 | 25,391 | 36.55 | 34.30 | 29.53 | 42.83 | 37.16 | 131 (112) | 86 (79) | 37 (29) | 8 (4) |
| *B. spectabilis* 'Splendens' | OR344372 | 154,520 | 85,688 | 18,078 | 25,377 | 36.46 | 34.18 | 29.47 | 42.81 | 37.18 | 131 (112) | 86 (79) | 37 (29) | 8 (4) |
| *× buttiana* 'Barbara Karst' | OR344369 | 158,231 | 85,688 | 21,789 | 25,377 | 36.34 | 34.18 | 29.81 | 42.81 | 37.18 | 131 (112) | 86 (79) | 37 (29) | 8 (4) |
| *B. × buttiana* 'San Diego Red' | OR344377 | 154,542 | 85,695 | 18,077 | 25,385 | 36.46 | 34.17 | 29.47 | 42.81 | 37.18 | 131 (112) | 86 (79) | 37 (29) | 8 (4) |
| *B. × buttiana* 'Miss Manila' sp. 1 | OR344378 | 154,520 | 85,688 | 18,078 | 25,377 | 36.46 | 34.18 | 29.47 | 42.81 | 37.18 | 131 (112) | 86 (79) | 37 (29) | 8 (4) |

Note: CDS, protein-coding gene; GC, guanine-cytosine; LSC, large single-copy region; SSC, small single-copy region; IR, inverted repeat.

transfer RNA (tRNA) and ribosomal RNA (rRNA) sequences were confirmed by tRNAscanSE v. 2.0.5 [23] and BLAST v. 2.13.0 [24]. The annotated complete chloroplast genome sequences were first validated using online GB2sequin [25], then verified and formatted using Sequin v. 15.50 from NCBI and deposited in GenBank (accession numbers are shown in Table 1). Chloroplast genome maps were drawn using Organellar Genome Draw (OGDRAW) v. 1.3.1 [26].

## Analyses of SSRs and long repeats

MIcroSAtellite (MISA) was used to identify simple sequence repeats (SSRs) in the thirteen newly sequenced *Bougainvillea* chloroplast genomes [27]. The parameters for di-, tri-, tetra-, penta-, and hexa-nucleotide SSRs and the minimum number of repeats were set to 10, 5, 4, 3, 3, and 3, respectively.

REPuter software [28] was used to identify and analyze the sizes and positions of long repeats, including forward, palindrome, reverse and complement repeat units, within the thirteen newly sequenced *Bougainvillea* chloroplast genomes. Long repeats were detected with a minimum repeat size of 30 bp, a Hamming distance of 3, and a repeat identity of more than 90%.

## Analysis of codon usage

The relative synonymous codon usage (RSCU) and amino acid frequencies of the thirteen newly sequenced *Bougainvillea* chloroplast genomes were analyzed using MEGA v. 7.0 [29]

with default parameters. A clustered heatmap of the RSCU values of the thirteen newly sequenced *Bougainvillea* chloroplast genomes was constructed with R v. 4.0.2 (https://www.R-project.org) (accessed on 10 August 2023).

## Comparative genomics and sequence divergence analyses

For comparison, 13 newly sequenced *Bougainvillea* chloroplast genomes were obtained using the CGView server [30]. GC contents were detected based on GC skew using the equation: GC skew = $(G - C)/(G + C)$. To further evaluate the variations among these 13 complete genomes of *Bougainvillea*, first, single-nucleotide polymorphisms (SNPs) and insertions/deletions (indels) were also identified and located using MUMmmer 4 [31] and Geneious Prime 2022.10 [32], using the annotated *B. glabra* 'White Stripe' as the reference; second, except *B. glabra* 'White Stripe', the rest 12 *Bougainvillea* cultivar chloroplast genomes were compared and analyzed to identify SNPs and indels using the annotated chloroplast genome of *B. × buttiana* 'Mahara' as the reference; third, to identify SNPs and indels between two complete chloroplast genomes of *B. spectabilis* 'Splendens', of which one was sequenced in this study and the other one was reported in a previous study [13], the reported one (OR253994) was used as the reference.

The mVISTA program in the Shuffle-LAGAN mode [33] and sliding window analysis using DnaSP v. 6.12.03 [34] were also employed to compare the complete chloroplast genome divergence among *Bougainvillea* cultivars. In total, 16 complete chloroplast genomes of *Bougainvillea* cultivars were analyzed, including 13 newly sequenced chloroplast genomes and 3 from the GenBank database (GenBank numbers MW557548, MW557549, and MW557550). The chloroplast genome of *B. × buttiana* 'Mahara' was used as the reference. Among these 16 chloroplast genomes of *Bougainvillea*, the LSC/IR and SSC/IR boundaries and their adjacent genes were also analyzed using IRscope [35].

## Selection pressure analysis in the Nyctaginaceae family

Selection pressure was applied following a previously described method [18]. In short, to detect positively selected amino acid sites among 46 complete chloroplast genomes of the Nyctaginaceae family (S2 Table), the nonsynonymous (dN) and synonymous (dS) substitution rates of consensus protein-coding genes were calculated by using the CodeML program implemented in EasyCodeML [36]. Gene selective pressure analysis was based on 79 consensus protein-coding gene sequences after removing all stop codons. The positive selection model of M8 (β & ω > 1) was used to detect positively selected sites based on both the dN and dS ratios (ω) and likelihood ratio test (LRT) values [37]. The Bayesian empirical Bayes (BEB) method was used to identify the codons most likely under positive selection, with posterior probabilities higher than 0.95 and 0.99 indicating sites under positive selection and strong positive selection, respectively [38].

## Phylogenetic relationships in the *Bougainvillea* genus and the Nyctaginaceae family

To reconstruct the phylogenetic relationships of the Nyctaginaceae family, 46 chloroplast genomes, including the 13 genomes generated in this study and 33 genomes downloaded from the GenBank database, were analyzed (S2 Table). *Seguieria aculeata* (NC_041418), *Rivina humilis* (NC_041300), *Petiveria alliacea* (NC_041417), and *Monococcus echinophorus* (NC_041414) were used as outgroups. Chloroplast genome sequences and protein-coding sequences were aligned using MAFFT v. 7.458 [39] with default parameters. Phylogenetic trees were constructed using the maximum likelihood (ML) and Bayesian inference (BI) methods. The best nucleotide substitution model (GTR + G + I) was determined using the Akaike

information criterion (AIC) in jModelTest v. 2.1.10 [40]. ML analysis was conducted in PhyML v. 3.0 [41] with 1000 bootstrap replicates. BI analysis was performed in MrBayes v. 3.2.6 [42]. Two Markov chain Monte Carlo (MCMC) algorithm runs were conducted simultaneously with 200,000 generations and four Markov chains, starting from random trees, sampling trees every 100 generations, and discarding the first 10% of samples as burn-in. The phylogenetic trees were edited and visualized using iTOL v. 3.4.3 (http://itol.embl.de/itol.cgi) (accessed on 15 September 2023).

## Results

### General characteristics of the thirteen complete chloroplast genomes

In this study, the 13 newly sequenced chloroplast genomes of *Bougainvillea* cultivars exhibited a typical quadripartite structure containing one large single-copy (LSC), one small single-copy (SSC) and two inverted-repeat regions (IRa and IRb) according to the OGDRAW and CGView tools (Fig 2, Table 1). The sizes of these 13 *Bougainvillea* chloroplast genomes ranged from 153,994 bp (*B. spectabilis* 'Flame') to 158,231 bp (*B.* × *buttiana* 'Barbara Karst') (Table 1). Among these 13 chloroplast genomes, four junction regions were identified, namely, one LSC region of 85,169–85,695 bp, one SSC region of 18,050–21,789 bp, and a pair of IR regions (IRa and IRb) of 25,377–25,426 bp each (Fig 2, Table 1). The GC contents of these 13 chloroplast

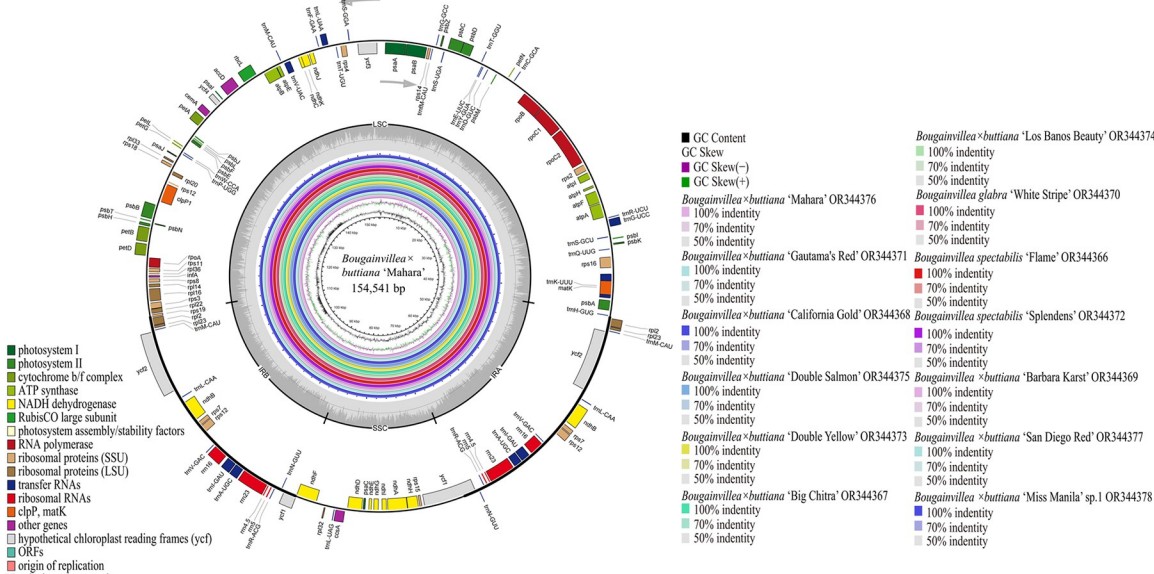

**Fig 2. Chloroplast genome map of *B.* × *buttiana* 'Mahara' (the outermost three rings) and CGView comparison of 13 complete chloroplast genomes of *Bougainvillea* cultivars (the inner rings with different colors).** Genes shown on the outside of the outermost first ring are transcribed counter-clockwise, and those on the inside are transcribed clockwise. The second ring with the darker gray color corresponds to the GC content, whereas the third ring with the lighter gray color corresponds to the AT content of the *B.* × *buttiana* 'Mahara' chloroplast genome generated by OGDRAW. The gray arrowheads indicate the directions of the genes. LSC, large single -copy region; IR, inverted repeat; SSC, small single-copy region. The innermost first black ring indicates the chloroplast genome size of *B.* × *buttiana* 'Mahara'. The innermost second and third rings indicate deviations in the GC content and GC skew, respectively, in the chloroplast genome of *C. barbatus*: GC skew + indicates G > C, and GC skew − indicates G < C. CGView comparison of the 13 complete chloroplast genomes of *Bougainvillea* cultivars displayed from the innermost 4th colored ring to the outer 16th ring: *B.* × *buttiana* 'Mahara', *B.* × *buttiana* 'Gautama's Red', *B.* × *buttiana* 'California Gold', *B.* × *buttiana* 'Double Salmon', *B.* × *buttiana* 'Double Yellow', *B.* × *buttiana* 'Big Chitra', *B.* × *buttiana* 'Los Banos Beauty', *B. glabra* 'White Stripe', *B. spectabilis* 'Flame', *B. spectabilis* 'Splendens', *B.* × *buttiana* 'Barbara Karst', *B.* × *buttiana* 'San Diego Red', and *B.* × *buttiana* 'Miss Manila' sp. 1, respectively. Chloroplast genome similar and highly divergent locations are represented by continuous and interrupted track lines, respectively.

genomes varied from 36.34% to 36.55% (Table 1). The IR region had the highest GC content (42.81–42.85%), followed by the LSC region (34.17–34.30%), while the SSC region had the lowest GC content (29.47–29.81%) (Table 1). The GC content of the protein-coding regions varied from 37.16% to 37.85%. All 13 *Bougainvillea* chloroplast genomes were submitted to the GenBank database (accession numbers OR344366–OR344378) (Table 1).

Among these 13 *Bougainvillea* chloroplast genomes, each contained 131 annotated functional genes, which consisted of 86 protein-coding genes, 37 transfer RNA (tRNA) genes, and 8 ribosomal RNA (rRNA) genes (Tables 1, 2 and S3). Among these genes, a total of 112 different genes were found in these 13 genomes, including 79 protein-coding genes, 29 tRNA genes, and 4 rRNA genes (Tables 1, 2 and S3). Overall, 17 genes contained introns in each of these 13 genomes. Fifteen genes (*atpF*, *ndhA*, *ndhB*, *petB*, *petD*, *rpl16*, *rpoC1*, *rps12*, *rps16*, *trnA-UGC*, *trnG-UCC*, *trnI-GAU*, *trnK-UUU*, *trnL-UAA*, and *trnV-UAC*) contained one intron, while *clpP* and *ycf3* each contained two introns (Tables 2 and S3). Among the 17 intron-containing genes in these 13 genomes, three genes (*ndhB*, *trnA-UGC* and *trnI-GAU*) occurred in both IRs; 12 genes (*atpF*, *clpP*, *petB*, *petD*, *rpl16*, *rpoC1*, *rps16*, *trnG-UCC*, *trnK-UUU*, *trnL-UAA*, *trnV-UAC* and *ycf3*) were distributed in the LSC; one gene (*ndhA*) was in the SSC; and one gene (*rps12*) in the first exon was located in the LSC, with the other two exons in both IRs (S3 Table).

**Table 2. Genes present in the 13 newly sequenced chloroplast genomes of *Bougainvillea* cultivars.**

| Gene category | Gene group | Gene names |
|---|---|---|
| **Self-replication** | DNA-dependent RNA polymerase | *rpoA*, *rpoB*, *rpoC1**, *rpoC2* |
| | Large subunit of ribosomal proteins | *rpl2* (×2), *rpl14*, *rpl16**, *rpl20*, *rpl22*, *rpl23* (×2), *rpl32*, *rpl33*, *rpl36* |
| | Small subunit of ribosomal proteins | *rps2*, *rps3*, *rps4*, *rps7* (×2), *rps8*, *rps11*, *rps12* (×2)*, *rps14*, *rps15*, *rps16**, *rps18*, *rps19* |
| **RNA genes** | Ribosomal RNA | *rrn4.5* (×2), *rrn5* (×2), *rrn16* (×2), *rrn23* (×2) |
| | Transfer RNA | *trnA-UGC* (×2)*, *trnC-GCA*, *trnD-GUC*, *trnE-UUC*, *trnF-GAA*, *trnfM-CAU*, *trnG-GCC*, *trnG-UCC**, *trnH-GUG*, *trnI-GAU* (×2)*, *trnK-UUU**, *trnL-CAA* (×2), *trnL-UAA**, *trnL-UAG*, *trnM-CAU* (×3), *trnN-GUU* (×2), *trnP-UGG*, *trnQ-UUG*, *trnR-ACG* (×2), *trnR-UCU*, *trnS-GCU*, *trnS-GGA*, *trnS-UGA*, *trnT-GGU*, *trnT-UGU*, *trnV-GAC* (×2), *trnV-UAC**, *trnW-CCA*, *trnY-GUA* |
| **Photosynthesis related genes** | Subunits of photosystem I | *psaA*, *psaB*, *psaC*, *psaI*, *psaJ* |
| | Subunits of photosystem II | *psbA*, *psbB*, *psbC*, *psbD*, *psbE*, *psbF*, *psbH*, *psbI*, *psbJ*, *psbK*, *psbL*, *psbM*, *psbN*, *psbT*, *psbZ*, *infA* |
| | Subunits of cytochrome b/f complex | *petA*, *petB**, *petD**, *petG*, *petL*, *petN* |
| | Subunits of ATP synthase | *atpA*, *atpB*, *atpE*, *atpF**, *atpH*, *atpI* |
| | Subunits of NADH dehydrogenase | *ndhA**, *ndhB* (×2)*, *ndhC*, *ndhD*, *ndhE*, *ndhF*, *ndhG*, *ndhH*, *ndhI*, *ndhJ*, *ndhK* |
| | Subunit of rubisco | *rbcL* |
| **Other genes** | Subunit of acetyl-CoA-carboxylase | *accD* |
| | c-type cytochrome synthesis gene | *ccsA* |
| | Envelope membrane protein | *cemA* |
| | Protease | *clpP*** |
| | Maturase | *matK* |
| **Genes of unknown function** | Conserved open reading frames | *ycf1* (×2), *ycf2* (×2), *ycf3***, *ycf4* |

Note: ×2: Gene with two copies; ×3: Gene with three copies; *: Gene containing only one intron; **: Gene containing two introns.

## SSRs and long repeats analyses

In the present study, the number of detected SSRs ranged from 86 (*B. spectabilis* 'Flame') to 98 (*B.* × *buttiana* 'Barbara Karst') among these 13 genomes (Fig 3A). Five types of SSRs were identified, including mononucleotide, dinucleotide, trinucleotide, tetranucleotide, and penta-nucleotide (Fig 3A, S4 Table). There were no hexanucleotides in any of the 13 sequenced genomes (Fig 3A). Among these 13 genomes, most SSRs were located in the LSC regions (67–76 loci) rather than in the SSC regions (13–14 loci) and IR regions (3 loci) (Fig 3B, S4 Table).

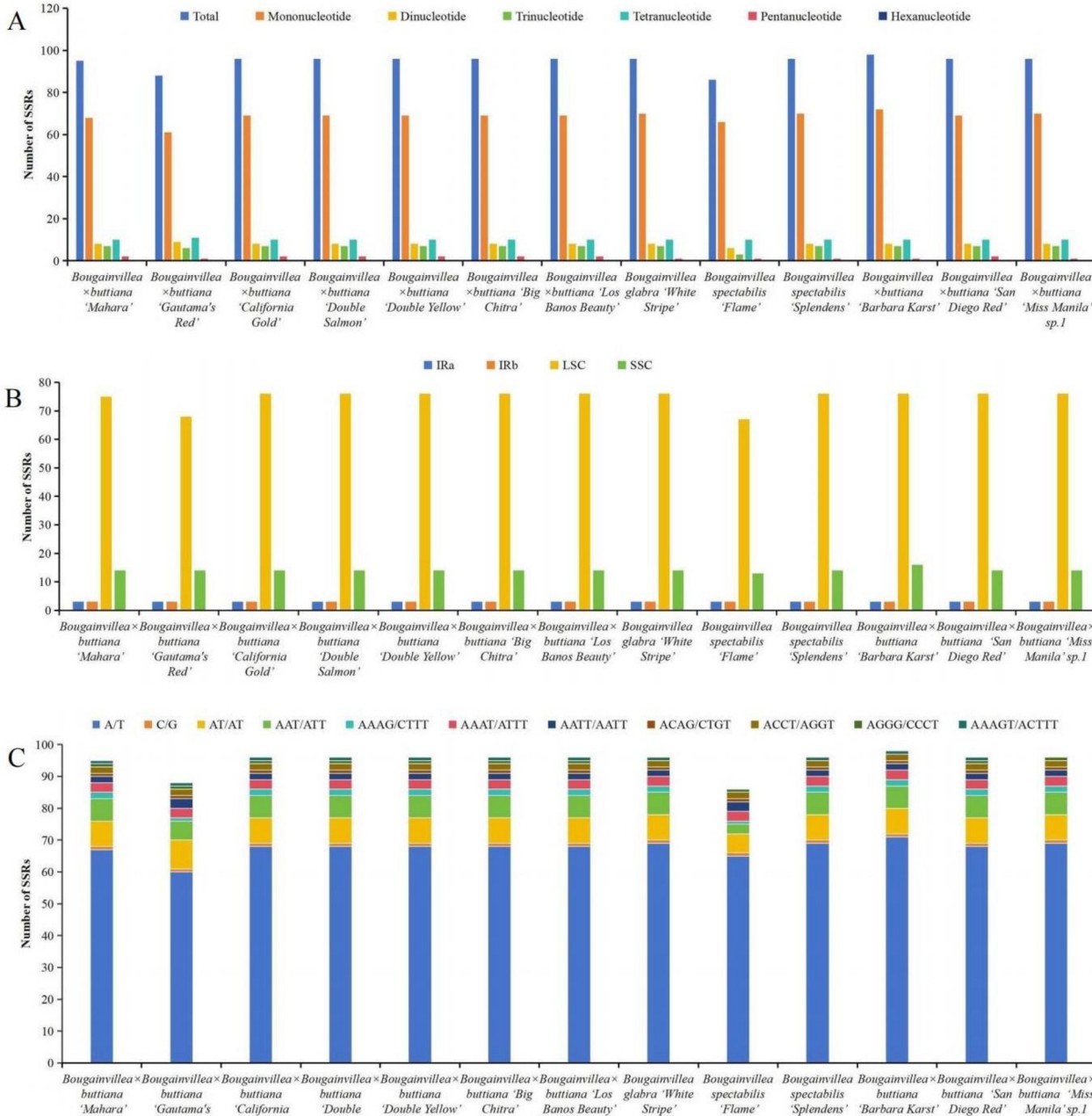

**Fig 3. Distribution of SSRs in the 13 newly sequenced *Bougainvillea* chloroplast genomes.** (A) Numbers of different SSR types detected in the 13 chloroplast genomes. (B) Frequencies of SSRs in the LSC, IR and SSC regions. (C) Frequencies of identified SSR motifs in different repeat class types.

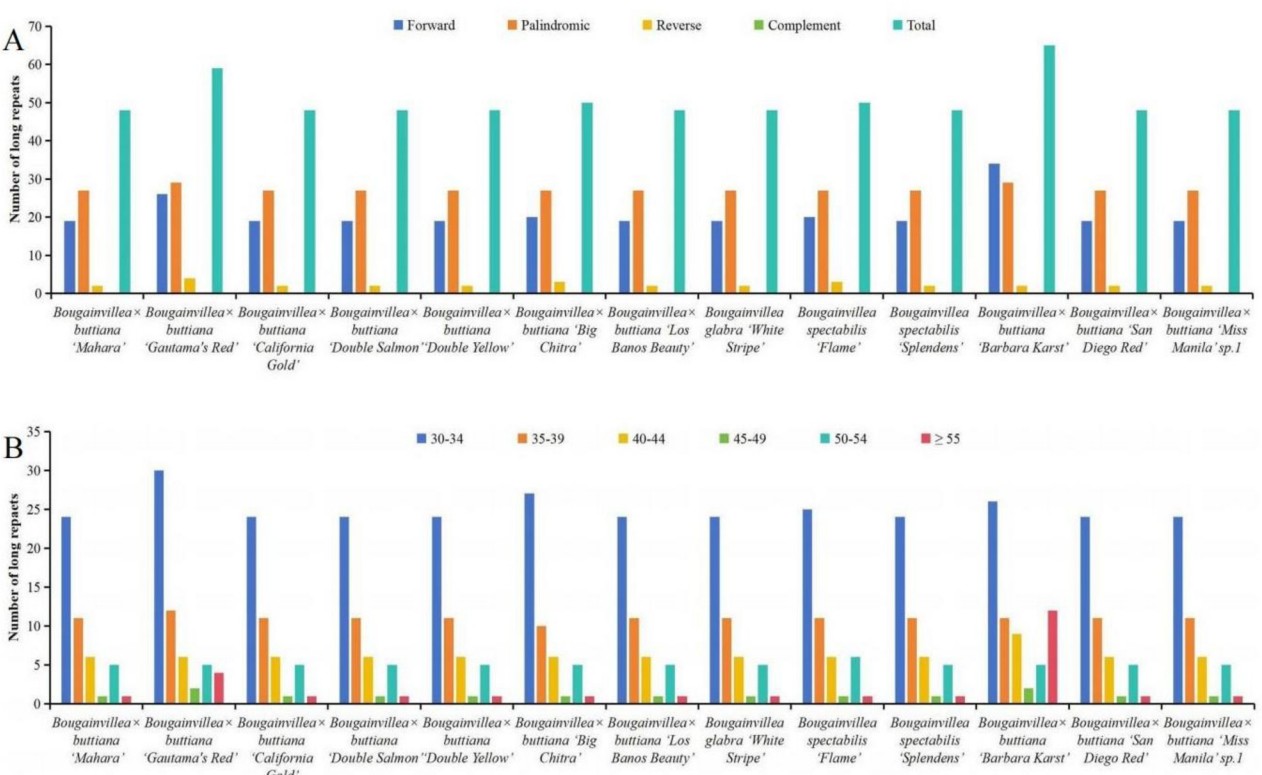

**Fig 4. Analysis of long repeat sequences in the 13 newly sequenced *Bougainvillea* chloroplast genomes.** (A) Total numbers of four long repeat types. (B) Length distribution of long repeats in each sequenced chloroplast genome.

Among each sequenced chloroplast genome, mononucleotide repeats were the most frequent, with numbers ranging from 61 to 72, followed by tetranucleotides ranging from 10 to 11, dinucleotides ranging from 6 to 9, trinucleotides ranging from 3 to 7, and pentanucleotides ranging from 1 to 2 (Fig 3C, S4 Table). Most of the mononucleotide SSRs were A/T repeats, which accounted for 68.18–75.58% of all SSRs among these 13 chloroplast genomes (Fig 3C, S4 Table). Among dinucleotide repeats, AT/AT repeats were observed most frequently, accounting for 6.98–10.22% of all SSRs (Fig 3C, S4 Table). In the trinucleotide category, AAT/ATT repeats were the most abundant type, accounting for 6.82–7.37% of all SSRs (Fig 3C, S4 Table).

Additionally, four different types of long repeats, including forward, complement, reverse, and palindromic repeats, were detected among these 13 chloroplast genomes. The total number of long repeats ranged from 48 to 65 (Fig 4A, S5 Table). The number of forward repeats varied from 19 to 34, the number of palindromic repeats varied from 27 to 29, and the number of reverse repeats varied from 2 to 4 (Fig 4A, S5 Table). There were no complement repeats in these 13 chloroplast genomes. The length of long repeats varied among these 13 chloroplast genomes (Fig 4B, S5 Table). Long repeats of 30–34 bp were found to be the most common, and those with lengths of 35–39 bp and 40–44 bp were the second and third most common, respectively (Fig 4B, S5 Table). These results indicated that the number, length and distribution of long repeats varied among these 13 chloroplast genomes in this study.

## Codon usage analysis

In this study, the codon usage, amino acid frequency, and relative synonymous codon usage (RSCU) of the 13 chloroplast genomes of Bougainvillea were analyzed (Fig 5). Methionine

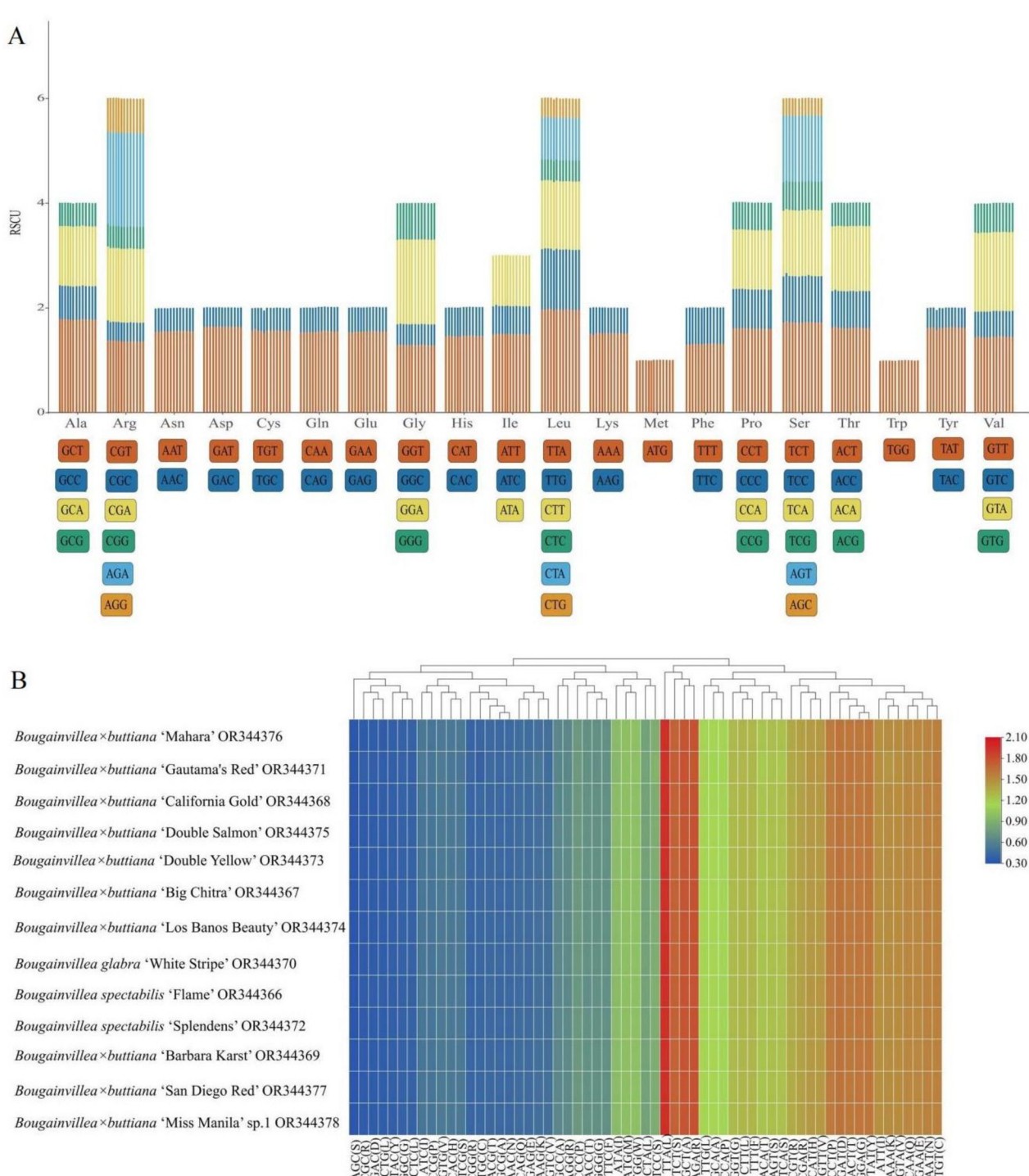

**Fig 5. Codon contents of all protein-coding genes of 13 newly sequenced complete chloroplast genomes of the *Bougainvillea* cultivars.** (A) Codon content and codon usage of the 20 amino acids and stop codons of all protein-coding genes. Each histogram from left to right is shown for *B. × buttiana* 'Mahara', *B. × buttiana* 'Gautama's Red', *B. × buttiana* 'California Gold', *B. × buttiana* 'Double Salmon', *B. × buttiana* 'Double Yellow', *B. × buttiana* 'Big Chitra', *B. × buttiana* 'Los Banos Beauty', *B. glabra* 'White Stripe', *B. spectabilis* 'Flame', *B. spectabilis* 'Splendens', *B.× buttiana* 'Barbara Karst', *B.× buttiana* 'San Diego Red', and *B. × buttiana* 'Miss Manila' sp. 1, respectively. (B) Heatmap analysis of the codon distribution of all protein-coding genes in the 13 newly sequenced chloroplast genomes.

(Met) and tryptophan (Trp) are each encoded by one codon, showing no codon bias, with RSCU values of 1.00, while the others are encoded by multiple synonymous codons, ranging from two to six (Fig 5A). The codons with the four lowest RSCU values (AGC, CGC, CTG and GAC) and four with the highest RSCU values (TTA, TCT, GCT and AGA) were identified in these 13 chloroplast genomes of *Bougainvillea* (Fig 5B). With the exception of Met and Trp, codon usage bias was detected for 29 codons with RSCU > 1.00 in the genes of these 13 chloroplast genomes (S6 Table). Interestingly, of the 29 codons, 28 were A/T-ending codons. Therefore, our RSCU results indicated that all 13 chloroplast genomes of *Bougainvillea* had a greater frequency of A/T-ending than G/C-ending codons.

## SNPs and indels analyses among the thirteen complete chloroplast genomes

Using the chloroplast genome of *B. glabra* 'White Stripe' as the reference, SNP/indel loci of the other 12 chloroplast genomes of *Bougainvillea* were detected (S1 Fig, Tables 3 and S7). Three comparisons, *B. glabra* 'White Stripe' vs. *B. spectabilis* 'Splendens', *B. glabra* 'White Stripe' vs. *B. × buttiana* 'Barbara Karst', and *B. glabra* 'White Stripe' vs. *B. × buttiana* 'Miss Manila' sp. 1, had no SNPs/indels. Five comparisons, *B. glabra* 'White Stripe' vs. *B. × buttiana* 'California Gold', *B. glabra* 'White Stripe' vs. *B. × buttiana* 'Double Salmon', *B. glabra* 'White Stripe' vs. *B. × buttiana* 'Double Yellow', *B. glabra* 'White Stripe' vs. *B. × buttiana* 'Los Banos Beauty', and *B. glabra* 'White Stripe' vs. *B. × buttiana* 'San Diego Red', identified the same numbers of SNPs

**Table 3. Distribution of SNPs and indels among the 13 newly sequenced complete chloroplast genomes of *Bougainvillea* cultivars.**

| Comparison pairs | Insertions | Deletions | Indels | Protein- coding genes SNPs | Intergenic regions SNPs | Total SNPs |
|---|---|---|---|---|---|---|
| 'White Stripe' vs. 'Big Chitra' | 1 | 2 | 3 | 1 | 6 | 7 |
| 'White Stripe' vs. 'California Gold' | 1 | 2 | 3 | 1 | 4 | 5 |
| 'White Stripe' vs. 'Double Salmon' | 1 | 2 | 3 | 1 | 4 | 5 |
| 'White Stripe' vs. 'Double Yellow' | 1 | 2 | 3 | 1 | 4 | 5 |
| 'White Stripe' vs. 'Gautama's Red' | 49 | 76 | 125 | 287 | 516 | 803 |
| 'White Stripe' vs. 'Los Banos Beauty' | 1 | 2 | 3 | 1 | 4 | 5 |
| 'White Stripe' vs. 'Mahara' | 1 | 3 | 4 | 1 | 8 | 9 |
| 'White Stripe' vs. 'San Diego Red' | 1 | 2 | 3 | 1 | 4 | 5 |
| 'White Stripe' vs. 'Flame' | 54 | 76 | 130 | 279 | 509 | 788 |
| 'Mahara' vs. 'Gautama's Red' | 50 | 74 | 124 | 284 | 492 | 776 |
| 'Mahara' vs. 'Big Chitra' | 0 | 0 | 0 | 0 | 2 | 2 |
| 'Mahara' vs. 'California Gold' | 0 | 0 | 0 | 0 | 2 | 2 |
| 'Mahara' vs. 'Double Salmon' | 0 | 0 | 0 | 0 | 2 | 2 |
| 'Mahara' vs. 'Double Yellow' | 0 | 0 | 0 | 0 | 2 | 2 |
| 'Mahara' vs. 'Los Banos Beauty' | 0 | 0 | 0 | 0 | 2 | 2 |
| 'Mahara' vs. 'Barbara Karst' | 4 | 1 | 5 | 1 | 6 | 7 |
| 'Mahara' vs. 'San Diego Red' | 0 | 0 | 0 | 0 | 2 | 2 |
| 'Mahara' vs. 'Miss Manila' sp. 1 | 3 | 1 | 4 | 1 | 6 | 7 |
| 'Mahara' vs. 'Flame' | 53 | 74 | 127 | 278 | 496 | 774 |
| 'Mahara' vs. 'Splendens' | 3 | 1 | 4 | 1 | 6 | 7 |

Note: 'White Stripe', 'Big Chitra', 'California Gold', 'Double Salmon', 'Double Yellow', 'Gautama's Red', 'Los Banos Beauty', 'Mahara', 'Barbara Karst','San Diego Red', 'Flame', 'Splendens' and 'Miss Manila' sp. 1 represent *B. glabra* 'White Stripe', *B.× buttiana* 'Big Chitra', *B.× buttiana* 'California Gold', *B.× buttiana* 'Double Salmon', *B.× buttiana* 'Double Yellow', *B.× buttiana* 'Gautama's Red', *B. × buttiana* 'Los Banos Beauty', *B.× buttiana* 'Mahara', *B.× buttiana* 'Barbara Karst', *B.× buttiana* 'San Diego Red', *B. spectabilis* 'Flame', *B. spectabilis* 'Splendens' and *B.× buttiana* 'Miss Manila' sp. 1, respectively.

and indels, with 1 protein-coding gene SNP, 4 intergenic SNPs, and 3 indels (S1 Fig, Tables 3 and S7). Two comparisons revealed slightly more SNPs and indels than did the other five comparisons. Regarding *B. glabra* 'White Stripe' vs. *B. × buttiana* 'Big Chitra', 1 protein-coding gene SNP, 6 intergenic SNPs, and 3 indels were identified; for *B. glabra* 'White Stripe' vs. *B. × buttiana* 'Mahara', 1 protein-coding gene SNP, 8 intergenic SNPs, and 4 indels were identified (S1 Fig, Tables 3 and S7). Between *B. glabra* 'White Stripe' and *B. × buttiana* 'Gautama's Red', 287 protein-coding gene SNPs, 516 intergenic SNPs, and 125 indels were detected (S1 Fig, Tables 3 and S7). With respect to *B. glabra* 'White Stripe' vs. *B. spectabilis* 'Flame', 279 protein-coding gene SNPs, 509 intergenic SNPs, and 130 indels were found (S1 Fig, Tables 3 and S7).

Except *B. glabra* 'White Stripe', the rest 12 *Bougainvillea* cultivar chloroplast genomes were also compared using the chloroplast genome of *B. × buttiana* 'Mahara' as the reference. Concerning *B. × buttiana* 'Mahara' vs. *B. × buttiana* 'Gautama's Red', 284 protein-coding gene SNPs, 492 intergenic SNPs, and 124 indels were found (S1 Fig, Tables 3 and S7). Six comparisons, *B. × buttiana* 'Mahara' vs. *B. × buttiana* 'Big Chitra', *B. × buttiana* 'Mahara' vs. *B. × buttiana* 'California Gold', *B. × buttiana* 'Mahara' vs. *B. × buttiana* 'Double Salmon', *B. × buttiana* 'Mahara' vs. *B. × buttiana* 'Double Yellow', *B. × buttiana* 'Mahara' vs. *B. × buttiana* 'Los Banos Beauty', and *B. × buttiana* 'Mahara' vs. *B.× buttiana* 'San Diego Red' had no indels (S1 Fig, Tables 3 and S7). However, these six comparisons had the same SNPs, with 2 intergenic SNPs each (S1 Fig, Tables 3 and S7). Interestingly, these 2 SNPs were both in *trnS-GCU_trnG-UCC-exon1* (S7 Table), suggesting that *trnS-GCU_trnG-UCC-exon1* can be used to identify these 7 *B. × buttiana* cultivars. Three comparisons, *B. × buttiana* 'Mahara' vs. *B. × buttiana* 'Barbara Karst', *B. × buttiana* 'Mahara' vs. *B. × buttiana* 'Miss Manila' sp. 1, and *B. × buttiana* 'Mahara' vs. *B. spectabilis* 'Splendens', had the same numbers of SNPs, with 1 protein-coding gene SNP and 6 intergenic SNPs. But these three comparisons identified different numbers of indels, with 5, 4, and 4 indels, respectively (S1 Fig, Tables 3 and S7). With respect to *B. × buttiana* 'Mahara' vs. *B. spectabilis* 'Flame', 278 protein-coding gene SNPs, 496 intergenic SNPs, and 127 indels were identified (S1 Fig, Tables 3 and S7).

## Intraspecific analyses of two chloroplast genomes of *B. spectabilis* 'Splendens'

The two chloroplast genomes from *B. spectabilis* 'Splendens' were found to show a 349 bp difference in length (OR344372 in Table 1 and OR253994 in [13]). With the total length difference, SNPs and indels were identified between the two complete chloroplast genomes of *B. spectabilis* 'Splendens'. Through intraspecific comparison, a total of 119 indels were identified between the two *B. spectabilis* 'Splendens' accessions (S8 Table). There were 55 insertions and 64 deletions between them (S8 Table). Among them, *cemA*, *rpl23*, *ycf1* and *ycf2* exhibited the same number indels, each of which showed 2 indels. There were 504 SNPs identified in the two complete chloroplast genomes of *B. spectabilis* 'Splendens' (S8 Table). The most frequently occurring mutations were G/T substitutions (72 times), followed by T/G (60 times), C/A (59 times), and T/C (57 times), respectively. Among these SNPs, *ycf1* contained the highest number of SNPs (36), followed by *rpoC2* and *ndhF*, which showed 11 and 9 SNPs, respectively (S8 Table).

## IR expansion and contraction

Comparisons of the LSC/IR and SSC/IR boundaries among these 13 chloroplast genomes and 3 published chloroplast genomes of *Bougainvillea* cultivars were performed (Fig 6). Regarding the LSC/IRb borders, the *rps19* gene was located at the boundaries of the LSC/IRb borders in all 16 *Bougainvillea* chloroplast genomes. The *rps19* gene expanded into the IRb region with a distance of 114 bp in all 16 *Bougainvillea* chloroplast genomes (Fig 6). Regarding the IRa/LSC

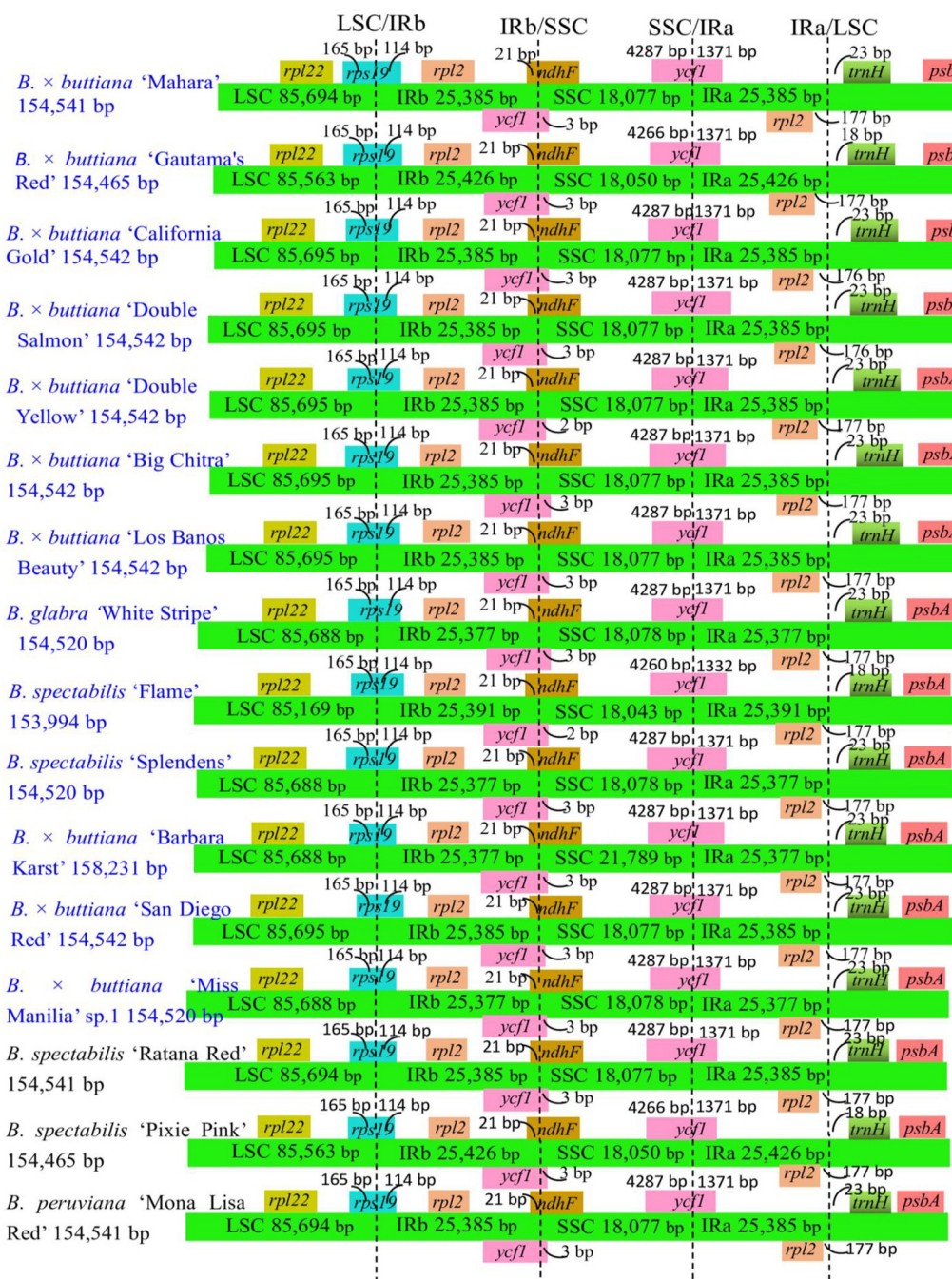

**Fig 6. Comparison of the borders of the LSC, SSC, and IR regions among the 16 *Bougainvillea* chloroplast genomes.**
The 13 newly sequenced *Bougainvillea* chloroplast genomes identified in this study are shown in blue.

borders, the *rpl2* and *trnH-GUG* genes were located at the boundaries of the IRa/LSC borders in all 16 *Bougainvillea* chloroplast genomes. The distances between the ends of the *rpl2* and IRa/LSC borders ranged from 176 bp to 177 bp (Fig 6). The distances between the ends of the *trnH-GUG* and IRa/LSC borders ranged from 18 bp to 23 bp (Fig 6).

The SSC/IRa border was located in the *ycf1* region, which crossed into the IRa region in all 16 *Bougainvillea* chloroplast genomes, with distances ranging from 1332 bp to 1371 bp (Fig 6).

For the IRb/SSC borders, *ycf1* expanded into the SSC regions in all 16 *Bougainvillea* chloroplast genomes and overlapped with the *ndhF* gene. A total distance of 2 or 3 bp was detected between the end of *ycf1* and the IRb/SSC border, and a 21 bp distance was detected between the start of *ndhF* and the IRb/SSC border (Fig 6). Overall, the LSC/IR boundary regions of the 16 *Bougainvillea* chloroplast genomes were highly conserved, but the SSC/IR boundary regions exhibited slight variations.

## Sequence divergence analysis

Multiple alignments of these 13 sequenced *Bougainvillea* chloroplast genomes were first compared by using CGView with the annotated genome sequence of *B.* × *buttiana* 'Mahara' as the reference (Fig 2). The CGView results indicated that no significant rearrangements were observed among these 13 chloroplast genomes, but several regions showed more divergence than others (the innermost 4th color ring to the outer 14th ring in Fig 2). Specifically, *trnT-GGU_psbD* and *trnT-GGU_trnE-UUC* in the LSC region were highly divergent.

To further detect sequence divergence in the chloroplast genomes of *Bougainvillea* cultivars, highly divergent regions in the 13 sequenced genomes in this study and 3 from the GenBank database were analyzed using mVISTA and DnaSP, with the annotated genome sequence of *B.* × *buttiana* 'Mahara' used as the reference (Fig 7). The mVISTA results showed that the LSC and SSC regions were more divergent than the two IR regions and that a greater divergence was found in the non-coding regions than in the coding regions (Fig 7). The main divergences for the coding regions were *psbJ*, *psaI*, and *ycf1*. For the non-coding regions, the strongly divergent regions were *trnH-GUG_psbA*, *psbI_trnS-GCU*, *trnS-GCU_trnG-UCC*, and *ccsA_ndhD* (Fig 7). For nucleotide diversity (Pi) values, the Pi values for the protein-coding regions ranged from 0 to 0.00990, with an average value of 0.00078 (S9 Table). Of these protein-coding regions, 7 regions (*psaI*, *psbJ*, *petG*, *clpP-exon1*, *rps19*, *ndhF*, and *ycf1*) exhibited remarkably high values (Pi > 0.0038; Fig 8A). For the intron and intergenic regions, the Pi values ranged from 0 to 0.02047, with an average of 0.00298 (S9 Table). Among these intron and intergenic regions, the 8 most divergent regions, *trnH-GUG_psbA*, *psbI_trnS-GCU*, *trnS-GCU_trnG-UCC-exon1*, *trnR-UCU_atpA*, *trnS-GGA_rps4*, *petD-exon2_rpoA*, *ccsA_ndhD*, and *ndhI_ndhA-exon2*, with Pi values ranging from 0.01130 to 0.02047, were identified (Fig 8B). Additionally, using a region length ≥ 200 bp and a Pi value ≥ 01130 for the selection of potential molecular markers, 4 regions were obtained: *trnH-GUG_psbA*, *trnS-GCU_trnG-UCC-exon1*, *trnS-GGA_rps4*, and *ccsA_ndhD* (S9 Table).

## Selection pressure analysis of the Nyctaginaceae family

The ratios (ω) of non-synonymousnonsynonymous (dN) to synonymous (dS) substitutions (dN/dS) for all 79 shared protein-coding genes were analyzed across 46 complete chloroplast genomes in the Nyctaginaceae family. These 46 genomes belonged to 9 genera of Nyctaginaceae, namely, *Bougainvillea*, *Belemia*, *Mirabilis*, *Nyctaginia*, *Boerhavia*, *Acleisanthes*, *Pisonia*, *Guapira* and *Salpianthus*. According to the M8 (β & ω > 1) model, a total of 9 protein-coding genes were under positive selection with a posterior probability greater than 0.95 according to the Bayes empirical Bayes (BEB) method (Table 4). Among these genes, *rps12* harbored the greatest number of positive amino acid sites (33), followed by *rbc L* (7), *ndhF* (6), *ycf2* (3), *rpoB* (2), *rpoC2* (2), *ndhI* (1), *psbT* (1), and *ycf3* (1) (Table 4).

## Phylogenetic relationships

Four phylogenetic trees were constructed using chloroplast genome sequences and protein-coding genes via the ML and BI methods, respectively (Figs 9 and S2). Four species of

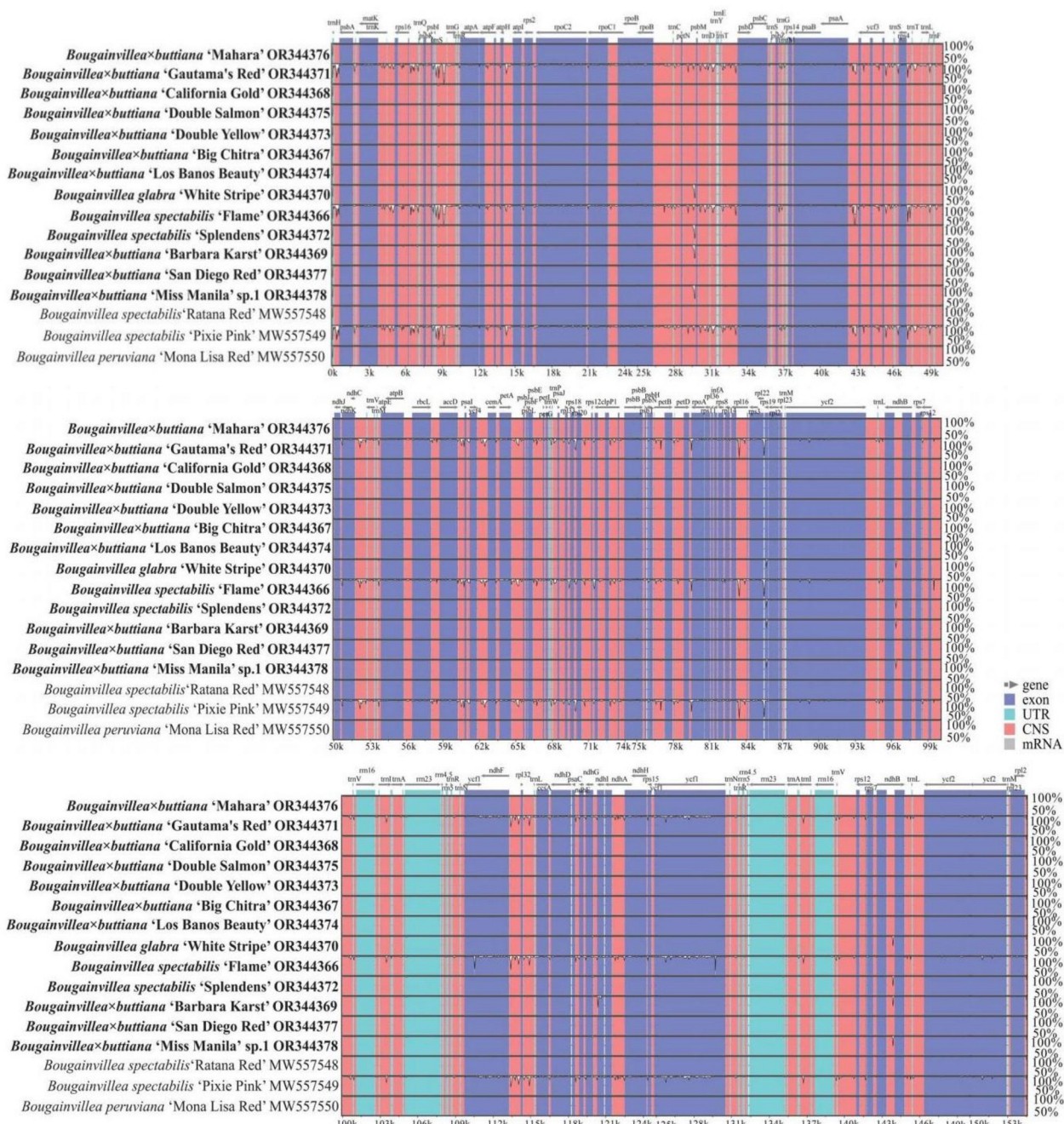

**Fig 7. Complete chloroplast genome comparison of the 16 *Bougainvillea* chloroplast genomes using *B. × buttiana* 'Mahara' as a reference.** The gray arrows and thick black lines above the alignment indicate gene orientation. Purple bars represent exons, sky-blue bars represent untranslated regions (UTRs), red bars represent non-coding sequences (CNS), gray bars represent mRNAs, and white regions represent sequence differences among the analyzed chloroplast genomes. The y-axis represents the identity percentage ranging from 50% to 100%. The 13 sequenced *Bougainvillea* chloroplast genomes in this study are shown in bold.

Petiveriaceae were used as outgroups. The ML and BI trees from complete chloroplast genomes and protein-coding genes showed similar topological structures within 9 genera of Nyctaginaceae and differed in support values and positions among several *Bougainvillea* cultivars (Figs 9 and S2). The 9 genera within Nyctaginaceae, *Bougainvillea*, *Belemia*, *Mirabilis*,

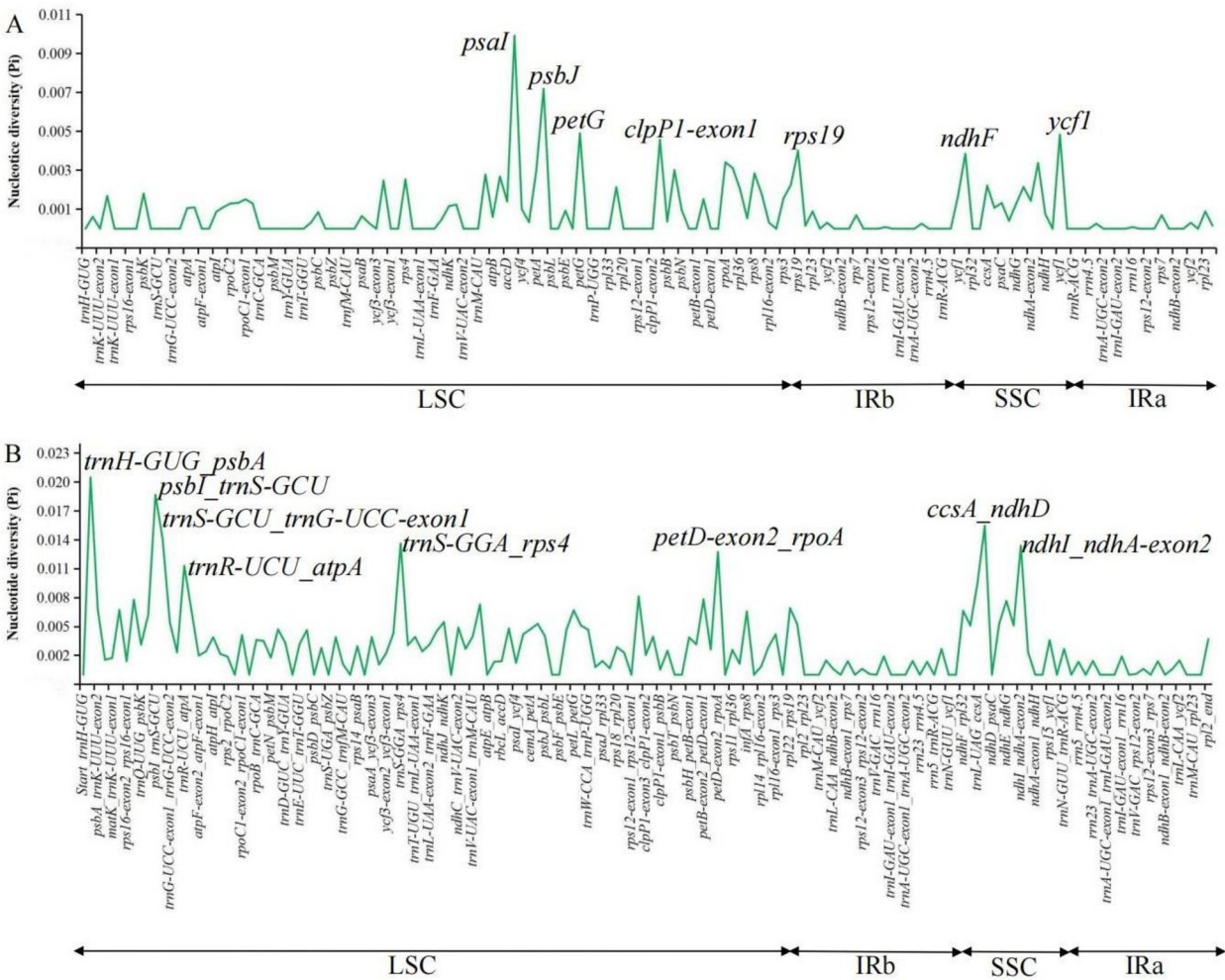

**Fig 8. Comparisons of nucleotide diversity (Pi) values among 16 complete chloroplast genomes of *Bougainvillea* cultivars.** (A) Protein-coding genes. Protein-coding genes with Pi values > 0.0038 are labeled with gene names. (B) Intergenic regions. Intergenic regions with Pi values > 0.0113 are labeled with intergenic region names.

*Nyctaginia*, *Boerhavia*, *Acleisanthes*, *Pisonia*, *Guapira* and *Salpianthus*, were strongly supported based on complete chloroplast genomes and protein-coding genes (bootstrap values, BS = 98–100% for the ML trees and posterior probabilities, PP = 1 for the BI trees) (Figs 9 and S2).

Within Nyctaginaceae, *Bougainvillea* was strongly supported as a sister to *Belemia* (BS = 98–99% for the ML trees and PP = 1 for the BI trees) (Figs 9 and S2). The 35 *Bougainvillea* individuals analyzed were divided into four clades, namely, Clades I, II, III, and IV, with strongly supported values (BS = 85–100% for the ML trees and PP = 0.99–1 for the BI trees) (Figs 9 and S2). Two cultivars, *B.* × *buttiana* 'Gautama's Red' and *B. spectabilis* 'Flame', were clustered into clade I, and the other 11 cultivars, including *B.* × *buttiana* 'Mahara', *B.* × *buttiana* 'California Gold', *B.* × *buttiana* 'Double Salmon', *B.* × *buttiana* 'Double Yellow', *B.* × *buttiana* 'Los Banos Beauty', *B.* × *buttiana* 'Big Chitra', *B.* × *buttiana* 'Barbara Karst', *B. glabra* 'White Stripe', *B. spectabilis* 'Splendens', *B.* × *buttiana* 'San Diego Red', and *B.* × *buttiana* 'Miss Manila' sp. 1, were clustered into Clade IV (Figs 9 and S2). In Clade I, *B. spectabilis* 'Flame'

**Table 4. Positively selected sites detected in 46 complete chloroplast genomes of the Nyctaginaceae family.**

| Gene | Positively selected sites (* $p > 95\%$; ** $p > 99\%$) |
|------|----------------------------------------------------------|
| *ndhF* | 462 L 0.969*, 502 A 0.960*, 508 T 0.955*, 518 F 0.958*, 573 L 0.980*, 576 Y 0.996** |
| *ndhI* | 166 E 0.999** |
| *psbT* | 34 M 0.982* |
| *rbcL* | 23 T 0.994**, 28 N 1.000**, 32 Q 0.959*, 225 L 0.998**, 359 N 0.994**, 439 R 0.995**, 477 K 1.000** |
| *rpoB* | 88 Q 0.990*, 363 W 0.951* |
| *rpoC2* | 556 L 0.992**, 706 Q 0.991** |
| *rps12* | 1 M 1.000**, 2 P 1.000**, 3 T 1.000**, 4 N 1.000**, 5 T 1.000**, 6 R 1.000**, 7 Q 1.000**, 8 P 1.000**, 9 I 1.000**, 10 K 1.000**, 11 N 1.000**, 12 V 1.000**, 13 T 1.000**, 14 K 1.000**, 15 S 1.000**, 16 P 1.000**, 17 A 1.000**, 18 L 1.000**, 19 R 1.000**, 20 G 1.000**, 21 C 1.000**, 22 P 1.000**, 23 Q 1.000**, 24 R 1.000**, 25 R 1.000**, 26 G 1.000**, 27 T 1.000**, 28 C 1.000**, 29 T 1.000**, 30 R 1.000**, 31 V 1.000**, 32 Y 1.000**, 110 K 0.997** |
| *ycf2* | 531 E 1.000**, 534 Y 0.999**, 1548 Q 0.999** |
| *ycf3* | 116 Q 0.997** |

Note: Each gene was assumed to have 95 degrees of freedom.

was sister to *B. peruviana* MW123901 and then formed a strong sister cluster to *B. pachyphylla*, both based on chloroplast genome sequences and protein-coding genes (Figs 9 and S2). However, the position of *B.* × *buttiana* 'Gautama's Red' in the ML tree constructed from chloroplast genome sequences differed from those in the other three phylogenetic trees in this study. For the former, *B.* × *buttiana* 'Gautama's Red' was sister to *B. spectabilis* 'Pixie Pink' and then clustered with *B. peruviana* MT407463 with strong support (BS = 90%) (Fig 9A). For the latter, *B. spectabilis* 'Pixie Pink' was sister to *B. peruviana* MT407463 and then clustered with *B.* × *buttiana* 'Gautama's Red' with strong support (BS = 93–100%, and PP = 0.93–1) (Figs 9 and S2). In Clade II, it only contained *B. spinosa* (Figs 9 and S2). In Clade III, there were 7 individuals of wild species, including *B. campanulata*, *B. berberidifolia*, *B. infesta*, *B. modesta* OM44398, *B. modesta* OM044396, *B. stipitata*, and *B. stipitata var. grisebachiana* (Figs 9 and S2). In Clade IV, in the ML tree based on the chloroplast genome sequences, *B.* × *buttiana* 'Mahara', *B.* × *buttiana* 'California Gold', *B.* × *buttiana* 'Double Salmon', *B.* × *buttiana* 'Double Yellow', *B.* × *buttiana* 'Los Banos Beauty', *B.* × *buttiana* 'Big Chitra', *B.* × *buttiana* 'San Diego Red', *B. spectabilis* 'Ratana Red', *B. glabra* MN888961, and *B. peruviana* 'Mona Lisa Red' were clustered together in one cluster with strong support (BS = 88–92%). *B.* × *buttiana* 'Barbara Karst', *B. glabra* 'White Stripe', *B. spectabilis* 'Splendens', *B.* × *buttiana* 'Miss Manila' sp. 1, *B. spectabilis* MN315508, *B. spectabilis* China MW167297, and *B.* hybrid cultivar MW123903 were clustered together in another cluster with strong support (BS = 88–95%) (Fig 9A). However, in the ML tree based on protein-coding genes, *B.* × *buttiana* 'Mahara' was sister to the other cultivars in Clade IV with strong support (BS = 100%) (S2A Fig). In both BI trees, *B.* × *buttiana* 'Mahara', *B.* × *buttiana* 'California Gold', *B.* × *buttiana* 'Double Salmon', *B.* × *buttiana* 'Double Yellow', *B.* × *buttiana* 'Los Banos Beauty', *B.* × *buttiana* 'Big Chitra', *B.* × *buttiana* 'San Diego Red', *B. spectabilis* 'Ratana Red', *B. glabra*, *B. peruviana* 'Mona Lisa Red', *B.* × *buttiana* 'Barbara Karst', *B. glabra* 'White Stripe', *B. spectabilis* 'Splendens', *B.* × *buttiana* 'Miss Manila' sp. 1, *B. spectabilis* MN315508, *B. spectabilis* MW167297, and *B.* hybrid cultivar MW123903 were clustered together in one cluster in Clade IV with moderate to strong support (PP = 0.84–1) (Figs 9B and S2B Fig). In the four phylogenetic trees, Clades III and IV were clustered together, forming a cluster with strong support (BS = 85–99%, and PP = 0.99–1); and then the cluster, Clade II, and Clade I were clustered step by step in the *Bougainvillea* genus with strong support (BS = 99–100%, and PP = 0.99–1) (Figs 9 and S2).

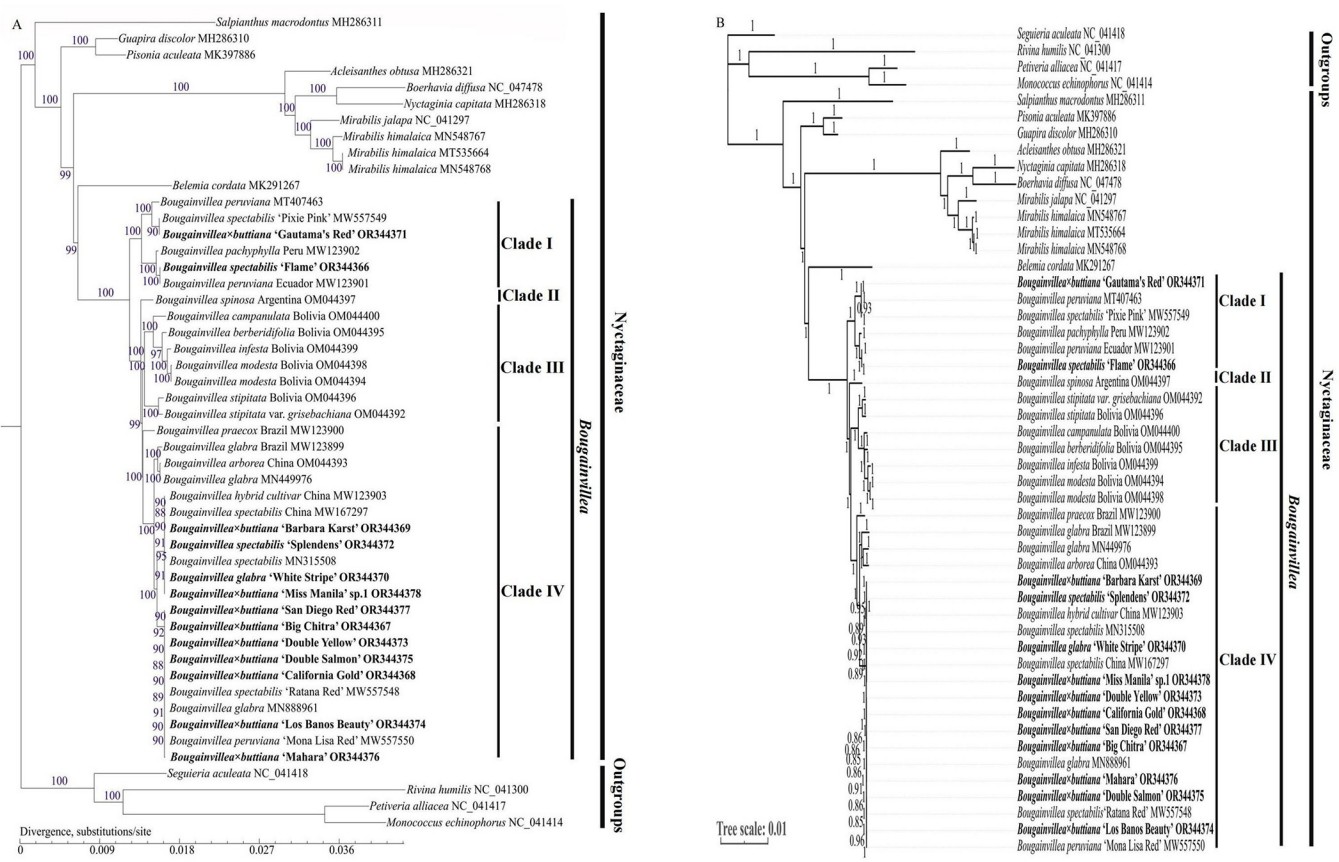

**Fig 9. Phylogenetic relationships of Nyctaginaceae species based on chloroplast genomes sequences reconstructed using maximum likelihood (ML) and Bayesian inference (BI) methods.** (A) ML tree. (B) BI tree. The 13 newly sequenced *Bougainvillea* chloroplast genomes identified in this study are shown in bold.

## Discussion

This study first analyzed the complete chloroplast genome sequences of ten *B.* × *buttiana* cultivars. Herein, all 13 sequenced chloroplast genomes possessed quadripartite structures, including one LSC, one SSC and two IR regions, and showed the same numbers of total genes, protein-coding genes, tRNA and rRNA genes, and introns, consistent with other reported chloroplast genomes of *Bougainvillea* [2,3,12,13]. There were some variations in the chloroplast genome lengths of these 13 cultivars, with *B.* × *buttiana* 'Barbara Karst' having the longest genome of 158,231 bp, and that of *B. spectabilis* 'Flame' having the shortest genome of 153,994 bp (Table 1). The genome sizes of the seven *B.* × *buttiana* cultivars in this study, *B.* × *buttiana* 'Mahara', *B.* × *buttiana* 'California Gold', *B.* × *buttiana* 'Double Salmon', *B.* × *buttiana* 'Double Yellow', *B.* × *buttiana* 'Los Banos Beauty', *B.* × *buttiana* 'Big Chitra', and *B.* × *buttiana* 'San Diego Red' were almost identical to the reported chloroplast genome of *B. glabra* (MN888961), which is 154,542 bp in length [11]. For *B.* × *buttiana* 'Gautama's Red', the genome size was the same as that of the chloroplast genome of *B. peruviana* (MT407463), which was 154,465 bp. The genome sizes of three cultivars, *B. spectabilis* 'Splendens', *B. glabra* 'White Stripe', and *B.* × *buttiana* 'Miss Manila' sp. 1, were the same as those of the reported chloroplast genome of *B. spectabilis* (MW167297), which is 154,520 bp [10]. Similar findings were also reported for the *Hyacinthus* and *Aglaonema* cultivars [8,9]. Among the three cultivars of *Hyacinthus*, 'Woodstock', 'Delft Blue' and 'Aiolos' had the same chloroplast genome

size of 154,640 bp [9]. Among the two *Aglaonema* cultivars, 'Hong Jian' and 'Red Valentine', also displayed the same genome size of 165,797 bp [8]. In the present study, seven *B. × buttiana* cultivars had the same chloroplast genome sizes, possibly because these chloroplast genomes did not undergo recombination through hybridization or grafting.

Because *B. × buttiana* cultivars are difficult to differentiate by their leaf appearance, developing molecular markers to identify them is important. In previous studies, highly divergent regions, SSRs, long repeats, SNPs, and indels were investigated among 20 wild species of *Bougainvillea* and one cultivar [2,3]. However, no studies on SNPs or indels among *B. × buttiana* cultivars have been previously reported. In the present study, 776 SNPs and 124 indels were found between *B. × buttiana* 'Mahara' and *B. × buttiana* 'Gautama's Red' (Tables 3 and S7). These SNPs and indels could be useful in the identification of these two *B. × buttiana* cultivars. Additionally, using *B. × buttiana* 'Mahara' as the reference, the other 6 *B. × buttiana* cultivars each had 2 intergenic SNPs (S1 Fig, Tables 3 and S7). These 2 SNPs were both in *trnS-GCU_trnG-UCC-exon1* (S7 Table). Therefore, *trnS-GCU_trnG-UCC-exon1* could be used to differentiate these 7 *B. × buttiana* cultivars. The other 9 comparisons, *B. glabra* 'White Stripe' vs. *B. × buttiana* 'California Gold', *B. glabra* 'White Stripe' vs. *B. × buttiana* 'Double Salmon', *B. glabra* 'White Stripe' vs. *B. × buttiana* 'Double Yellow', *B. glabra* 'White Stripe' vs. *B. × buttiana* 'Los Banos Beauty', *B. glabra* 'White Stripe' vs. *B. × buttiana* 'San Diego Red', *B. glabra* 'White Stripe' vs. *B. × buttiana* 'Big Chitra', *B. glabra* 'White Stripe' vs. *B. × buttiana* 'Mahara', *B. glabra* 'White Stripe' vs. *B. × buttiana* 'Gautama's Red', and *B. glabra* 'White Stripe' vs. *B. spectabilis* 'Flame', also contained SNPs and indels (Tables 3 and S7). These SNPs and indels could be used to identify these 10 cultivars. However, 3 comparisons, *B. glabra* 'White Stripe' vs. *B. spectabilis* 'Splendens', *B. glabra* 'White Stripe' vs. *B. × buttiana* 'Barbara Karst', and *B. glabra* 'White Stripe' vs. *B. × buttiana* 'Miss Manila' sp. 1 had no SNPs/indels. These 3 comparisons indicated that the chloroplast genomes of these 4 *Bougainvillea* cultivars did not undergo recombination during hybridization or grafting. The leaf color of *B. glabra* 'White Stripe' was yellow–green with white spots, while the leaf colors of *B. spectabilis* 'Splendens', *B. × buttiana* 'Barbara Karst' and *B. × buttiana* 'Miss Manila' sp. 1 were dark green (Fig 1). The bract color of *B. glabra* 'White Stripe' was white, while the bract colors of the three cultivars were red (Fig 1). The molecular regulatory mechanisms of leaf and bract color variations in these 4 cultivars need further study.

Highly divergent regions can be used as potential DNA markers for studies on phylogenetic relationships and species identification [43,44]. However, for some Nyctaginaceae species, phylogenetic relationships determined using universal DNA markers include multiple poor-resolution branches [1,6]. For example, three chloroplast DNA markers, namely, *ndhF*, *rps16*, and *rpl16*, and one nuclear *ITS* could not be used to identify *Acleisanthes lanceolatus* and *A. longiflora* [1]. Additionally, based on the Pi values studied here, it was also obvious that frequently used chloroplast markers, including *ndhF*, *rps16*, and *rpl16*, exhibited low polymorphism (0.0038, 0, and 0.0003, respectively) at the genus level in *Bougainvillea*. Therefore, it will be necessary to explore more highly divergent regions that represent potential markers for future studies. Currently, based on the results of mVISTA and nucleotide diversity analyses, 4 divergent regions among the 16 chloroplast genomes of the *Bougainvillea* cultivars in this study have been identified, including *trnH-GUG_psbA*, *trnS-GCU_trnG-UCC-exon1*, *trnS-GGA_rps4*, and *ccsA_ndhD* (Figs 2, 7 and 8). In comparison, *trnH-GUG_psbA* was also reported in wild *Bougainvillea* species and cultivars [2,13]; *ccsA-ndhD* was reported as a potential molecular marker in *Amomum* [45] and *Alpinia* [46]; and *trnS_trnG* was reported in *Kaempferia* [47]. Hence, based on these results, we suggest that these four divergent regions can be used as potential marker resources for *Bougainvillea* cultivar identification and phylogenetic analysis.

In this study, the ω ratio (ω = dN/dS) was used for measuring selective pressure in the Nyctaginaceae family. For most of the protein-coding genes, the value of ω was less than one, revealing that they were under purifying selection. Additionally, 9 genes, namely, *rps12*, *rbcL*, *ndhF*, *rpoB*, *rpoC2*, *ndhI*, *psbT*, *ycf2*, and *ycf3*, were identified as having positive selection sites in the Nyctaginaceae family (Table 3). Recent studies have indicated that these 9 genes are commonly undergoing positive selection in higher plants [43,48–54]. For example, *rpoC2*, *rps12*, *rbcL*, and *ycf2* have also been identified as being under positive selection in orchid species [43]; *rbcL*, *rpoC2*, *rps12*, and *ycf2* have been reported as being under positive selection in some Zingiberaceae species [48]; *ndhF*, *rbcL*, *rpoC2*, *rps12* and *ycf2* have also been identified as being under positive selection in Papilionoideae species [49]; *rpoC2*, *rps12*, *rbcL*, and *ycf3* have also been identified as being under positive selection in *Zingiber* [50,51]; *rpoB* and *rps12* have also been identified as being under positive selection in *Begonia* [52]; *rbcL* and *ycf2* have also been identified as being under positive selection in Monsteroideae [53]; and *ndhI* has also been identified as being under positive selection in *Saxifraga* [54]. The analyzed species of the Nyctaginaceae family possess diverse morphological and ecological characteristics; for instance, some species are distributed in the tropics, while other species are distributed in the subtropics; some species are high-elevation trees, while other species are low-elevation shrubs and herbs [1,2]. In other words, Nyctaginaceae species live in diverse habitats and have high levels of plant diversity. Therefore, Nyctaginaceae species may face different types of stresses in their ecological habitats, and these 9 positively selected genes may play important roles in the evolution and adaptation of Nyctaginaceae species to their respective ecological habitats.

In the present study, our four phylogenetic trees obtained from chloroplast genome sequences and protein-coding genes revealed that *Bougainvillea* was a sister to *Belemia* with strong support (BS = 98–99%, and PP = 1) (Figs 9 and S2). This result was broadly consistent with those of previous studies [1,6]. We also found that 35 *Bougainvillea* individuals within the *Bougainvillea* genus were divided into four clades with strong support (Figs 9 and S2). This finding was in agreement with a previous study [3], but it had difference with recently reported study [13]. For the former, *B. spinosa* was sister to clade II (the 'cultivated' *Bougainvillea* clade) or clade III (the 'wild' *Bougainvillea* clade) based on protein-coding genes of chloroplast genomes [3], whereas for the latter, the 19 *Bougainvillea* plants were clustered into 3 clades based on complete chloroplast genomes, and *B. spinosa* was classified into the third clade [13]. This might because the latter study did not use plenty of *Bougainvillea* samples for phylogenetic analysis. However, the four phylogenetic trees in this study displayed some inconsistencies in the *Bougainvillea* genus, such as the shifting position of *B.* × *buttiana* 'Gautama's Red' in Clade I (Figs 9 and S2). Therefore, more *Bougainvillea* cultivar chloroplast genomes may need to be sequenced to resolve their positions. Nonetheless, based on our phylogenetic results, we propose that *B.* × *buttiana* 'Gautama's Red' may be from *B. peruviana* and *B. spectabilis*. Additionally, the other eleven cultivars, including the remaining nine *B.* × *buttiana* cultivars, were clustered in Clade IV in the *Bougainvillea* genus (Figs 9 and S2). From the SNPs/indels analyses, seven of these nine *B.* × *buttiana* cultivars had no indels and only 2 SNPs (Table 3). Surprisingly, between *B.* × *buttiana* 'Mahara' and *B. spectabilis* 'Splendens', there existed relatively low numbers of SNPs/indels, with only 1 protein-coding gene SNP, 6 intergenic SNPs and 4 indels (S1 Fig, Tables 3 and S7). These results indicated that these analyzed *B.* × *buttiana* cultivars and *B. spectabilis* 'Splendens' showed close relationships. Considering that grafting techniques are often used in the cultivation processes of *B.* × *buttiana* cultivars, and based on the results of our four phylogenetic trees and SNPs/indels, we speculated that these nine *B.* × *buttiana* cultivars may come from hybrids involving *B. peruviana*, *B. spectabilis* and *B. glabra*. These results were, to some extent, in agreement with a previous hypothesis,

which presumed that the *B. × buttiana* cultivars may be hybrids of *B. peruviana* and *B. glabra* [5].

## Conclusions

In this study, 13 complete chloroplast genomes of 13 *Bougainvillea* cultivars from South China were sequenced, assembled and compared for genome structural characteristics. Furthermore, the molecular evolution of chloroplast genomes in the Nyctaginaceae family was studied, and the phylogenetic relationships of the Nyctaginaceae family, including *Bougainvillea* cultivars, were reconstructed with high-resolution branches. The 13 newly sequenced chloroplast genomes had a typical quadripartite structure, and each contained 112 different genes, including 79 protein-coding genes, 29 tRNA genes and 4 rRNA genes, with a chloroplast genome length of 153,994–158,231 bp. Comparative analyses of *Bougainvillea* cultivar chloroplast genomes revealed 4 highly divergent regions that can be used as potential markers for phylogenetic analyses and cultivar identification. Among the 46 chloroplast genomes of the Nyctaginaceae family, 9 protein-coding genes, namely, *rps12*, *rbcL*, *ndhF*, *rpoB*, *rpoC2*, *ndhI*, *psbT*, *ycf2*, and *ycf3*, were found to be undergoing positive selection at the amino acid site level. Based on complete chloroplast genomes and protein-coding genes, phylogenetic trees divided the *Bougainvillea* species and cultivars into 4 clades with strong support. These assembled chloroplast genomes enrich genomic resources and will help with the identification and utilization of *Bougainvillea* germplasm resources.

## Supporting information

**S1 Fig. Indels statistics of 13 newly sequenced complete chloroplast genomes of the *Bougainvillea* cultivars.** First, the *B. glabra* 'White Stripe' chloroplast genome was used as the reference sequence for indels analyses for the other twelve *Bougainvillea* chloroplast genomes. Second, except *B. glabra* 'White Stripe', the rest 12 *Bougainvillea* cultivar chloroplast genomes were compared using the chloroplast genome of *B. × buttiana* 'Mahara' as the reference. (A) Total indels statistics. (B) Insertion statistics. (C) Deletion statistics.
(DOCX)

**S2 Fig. Phylogenetic relationships of Nyctaginaceae species based on protein-coding genes reconstructed using ML and BI methods.** (A) ML tree. (B) BI tree. The 13 newly sequenced *Bougainvillea* chloroplast genomes identified in this study are shown in bold.
(DOCX)

**S1 Table. Information on the 13 *Bougainvillea* cultivars.**
(DOCX)

**S2 Table. The 46 complete chloroplast genomes in the Nyctaginaceae family used for determining the selective pressure and phylogenetic relationships.**
(DOCX)

**S3 Table. Genes distribution in the 13 chloroplast genomes of the *Bougainvillea* cultivars.**
(XLSX)

**S4 Table. SSRs detected in the 13 chloroplast genomes of the *Bougainvillea* cultivars.**
(XLSX)

**S5 Table. Long repeats detected in the 13 chloroplast genomes of the *Bougainvillea* cultivars.**
(XLSX)

**S6 Table. Codon usage in the 13 chloroplast genomes of the *Bougainvillea* cultivars.**
(XLSX)

**S7 Table. SNPs and indels detection among the 13 chloroplast genomes of the *Bougainvillea* cultivars.**
(XLSX)

**S8 Table. SNPs and indels detection between the 2 chloroplast genomes of *B. spectabilis* 'Splendens'.**
(XLSX)

**S9 Table. Nuclear diversity of 16 chloroplast genomes from the 16 *Bougainvillea* cultivars.**
(XLSX)

## Author Contributions

**Conceptualization:** Xiao-Ye Wu, Lan Wang, Dong-Mei Li.

**Data curation:** Dong-Mei Li.

**Formal analysis:** Xiao-Ye Wu, He-Fa Wang, Shui-Ping Zou, Lan Wang, Gen-Fa Zhu, Dong-Mei Li.

**Funding acquisition:** Shui-Ping Zou, Gen-Fa Zhu.

**Investigation:** Xiao-Ye Wu, He-Fa Wang, Lan Wang.

**Methodology:** Dong-Mei Li.

**Project administration:** Shui-Ping Zou, Gen-Fa Zhu.

**Resources:** Xiao-Ye Wu, He-Fa Wang, Lan Wang.

**Software:** Dong-Mei Li.

**Supervision:** Gen-Fa Zhu.

**Validation:** Xiao-Ye Wu, He-Fa Wang, Shui-Ping Zou, Lan Wang, Gen-Fa Zhu, Dong-Mei Li.

**Visualization:** Xiao-Ye Wu, He-Fa Wang, Shui-Ping Zou.

**Writing – original draft:** Dong-Mei Li.

**Writing – review & editing:** Dong-Mei Li.

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
