## [Decision Letter · Decision Letter 0]

4 Mar 2024

PONE-D-23-43492Comparative analysis of the complete chloroplast genomes of thirteen Bougainvillea cultivars from South China with implications for their genome structures and phylogenetic relationshipsPLOS ONE

Dear Dr. Li,

Thank you for submitting your manuscript to PLOS ONE. After careful consideration, we feel that it has merit but does not fully meet PLOS ONE’s publication criteria as it currently stands. Therefore, we invite you to submit a revised version of the manuscript that addresses the points raised during the review process.

 Please carefully address each concern raised by one of the reviewer, providing a point-by-point response to all the concerns. 

We look forward to receiving your revised manuscript.

Kind regards,

Pankaj Bhardwaj, Ph.D.

Academic Editor

PLOS ONE

Journal Requirements:

This research was financially supported by the Collection, Identification and Utilization of new and superior flower germplasm resources (2023), and Guangdong Province Modern Agriculture Industry Technical System-Flower Innovation Team Construction Project (2023KJ121). 

Reviewers' comments:

Reviewer's Responses to Questions

**Comments to the Author**

1. Is the manuscript technically sound, and do the data support the conclusions?

Reviewer #1: Partly

Reviewer #2: Yes

2. Has the statistical analysis been performed appropriately and rigorously? 

Reviewer #1: Yes

Reviewer #2: Yes

3. Have the authors made all data underlying the findings in their manuscript fully available?

Reviewer #1: Yes

Reviewer #2: Yes

4. Is the manuscript presented in an intelligible fashion and written in standard English?

Reviewer #1: No

Reviewer #2: Yes

5. Review Comments to the Author

Reviewer #1: The study of Wu et al. domenstrates the application of chloroplast genomes in explore the germplasm of Bougainvillea cultivars. Apart from the phylogenetic relationships between the cultivars, potential molecular markers including SSRs, LSRs and hotspot regions have been investigated from the chloroplast genomes. The study shows merits on the research of Bougainvillea, that could serve as a reference in studying other horticultural crops with diversified morphologies. However, prior to a make a further decision, the following major and minor issues should be resolved.

Major issues:

1) The cultivars of Bougainvillea are regulated by the International Code for the Nomenclature for Cultivated Plants (ICNCP). The registration of Bougainvillea cultivars is designated to the Indian Agricultural Research Institute (IARI). The authors are responsible to carefully check if the cultivar epithets are well established and registered. Illegitimate epithets (e.g. misspelling and homonyms) and unestablished epithets are common in horticultural germplasms. The authors should refer to the Articles 25 to 27 of ICNCP (9th Edition) which is available online (https://www.ishs.org/sites/default/files/static/ScriptaHorticulturae_18.pdf). The publications of The Bougainvillea Society of India (BSI) of IARI (http://www.bsi-iari.com/publication.htm) will help the authors in checking. The authors should indicate those unestablished and unregistered cultivars in the manuscript as precaution for the readers.

2) I wonder if the mutant (Bougainvillea sp.1) is discovered by the authors themselves? If so, what is the original cultivar of this mutant? The authors should elaborate why this mutant is included in this study. Also, Figure S1 (the images of 13 cultivars) should be included in the main text, prior to the chloroplast genome map. Vouchers of the studied cultivars, either live and dried specimens, should be listed out with their collection location, GPS, date of collection, collector numbers, and deposited herbarium or institution.

3) I noticed that Bougainvillea spinosa was included in Clade II by the authors. However, it should not be regarded as a member of Clade II, as it is sister to all other members from both clade II and III (in both Figure 8 A&B and S3 A&B). The authors should discuss the potential reasons in the Discussion.

4) The definition of SNPs and InDels in this manuscript should be well defined. Could the nucleotide differences between one and another cultivar be considered as SNPs and InDels? Molecular Diagnostic Characters (MDCs) of a cultivar to differentiate itself from the other 12 cultivars could be more meaningful. Also, haplotype analysis could aid in visualizing the figures in this tables by grouping the cultivars. The authors could refer to the following article:

Wong, K. H., Wu, H. Y., Kong, B. L. H., But, G. W. C., Siu, T. Y., Hui, J. H. L., Shaw, P. C., & Lau, D. T. W. (2022). Characterisation of the complete chloroplast genomes of seven Hyacinthus orientalis L. cultivars: Insights into cultivar phylogeny. Horticulturae, 8(5), 1-28.

Minor issues:

The writing style and language of the manuscript, particularly the abstracts, introduction and the discussion, should be meticulously improved. I have identified a number of mistakes in grammar and use of wordings which deteriorate the comprehensibility of the manuscript. In addition, I highly recommend the authors to employ a native English speaker or professional editing agency to proofread the manuscript. The style in presenting figures and tables should also be improved. Please kindly refer to the following:

L16-18: Having high similarity in leaf appearance and hybridization among Bougainvillea species, the phylogenetic relationships of the genus are complicated and controversial.

L20-21: Their phylogenetic relationships within the genus Bougainvillea and other species of the family Nyctaginaceae are identified for the first time.

L27-28: Four divergent regions, including ......, were identified from sliding window analysis of 16 Bougainvillea cultivar genomes.

L36: replace "which contained" by "including"

L39: ..., but also helped to identify Bougainvillea

L51: "Le Du Juan", which is well-known in China, belongs to the genus Bougainvillea of the family Nyctaginaceae.

L53-54: with colored bracts [2,3]. The colorful bracts surrounding the small tubular flowers are often mistakenly treated as flowers.

L59: remove "value"

L60: replace "only" by "mainly"

L60-61: ... challenging because of high similarity

L68: replace "identified by" by "explored using"

L69: Why is this sentence concerning the usage of Bougainvillea is placed here? Recommend placing it in the previous paragraphs.

L70: delete "that has been"

L71: replace "taken" by "introduce"

L75: ... B. x buttiana cultivars and the molecular evolution ...

L77-78: In this study, complete chloroplast genomes of thirteen Bougaivillea cultivars were newly sequence, assembled and annotated. These thirteen cultivars from South China include seven ....

L84: ... integrating three published ...

L85: Bougainvillea cultivars on the NCBI. In this study, five objectives were targeted: (1) ... [Aim =/= objectives!]

L97: please state the full name of the "resource garden" here.

L99: remove "finally"

L101-102: ... the DNA quality and quantity was checked through ... method, respectively. ...

L104: is the read in 150 bp paired or not???

L109: assembly and annotation were conducted using the methods previously reported [15].

L125: ... The long repeats were detected with a minimum ....

L143: remove "ones"

L163: Phylogenetic trees were constructed using ...

L186 and elsewhere throughout the manuscript: "different genes" should be better replaced as "unique genes".

L253 and the parts reporting IR: please keep consistency with the border titles with Fig. 5. For example, in L253 "IRb/LSC" should be replaced as "LSC/IRb".

L303 and elsewhere throughout the manuscript: the authority of genera and species should be shown for the first appearance in the manuscript. For example, Bougainvillea Comm. ex Juss., Belemia Pires, Mirabilis L. Latin should be italic.

L310-311: the authors could discuss more concerning the different phylogentic positions of the cultivars of B x buttiana 'Gautama's Red' and 'Mahara'. Would it suggest a different origin of them?

L325: Why B. glabra 'White Stripe' is not clustered with other cultivars of B. glabra?

L374: replace "with" by "in"

Fig 1. It might be meaningless to include the GCView Comparison results since they are too thin which are hardly can see. The layout of chloroplast genome map would be too complicated. The authors could consider removing the GC comparison from this figure and submit it as a supplementary figure.

Table 4: The authors should define what the numbers and alphabets represent in the table. It is recommended to rearrange the table in a more easy-understanding manner e.g. by dividing the items of a positively selected sites in columns:

Positively selected sites

Gene xx |x| xxx

ndhF 462 |L|0.969*

502 |A|0.960*

...

Reviewer #2: Dear authors

I have reviewed this study from the beginning to the end and I did not see any deficiency in terms of the main headings (introduction, material-method, conclusion and discussion). The methods were selected in accordance with the purpose of the study and applied correctly. The language of the article is simple and understandable. There are no stylistic errors. The main objectives of the study are: (1) to characterise and study the structures of 13 newly sequenced complete chloroplast genomes; (2) to detect variation in simple sequence repeats 87 (SSR), long repeats and codon usage among these 13 chloroplast genomes; (3) to identify highly variable 88 regions for potential development of DNA markers among Bougainvillea cultivars; (4) to understand the molecular 89 evolution of chloroplast genomes in the Nyctaginaceae family; and (5) to reveal the phylogenetic relationships 90 between Bougainvillea species and varieties and other species in the Nyctaginaceae family. Therefore, I would like to say that the study is acceptable in its present form. Sincerely.

6. PLOS authors have the option to publish the peer review history of their article (what does this mean?). If published, this will include your full peer review and any attached files.

Reviewer #1: **Yes: **Kwan-Ho WONG

Reviewer #2: No

---

## [Author Response · Author response to Decision Letter 0]

11 Apr 2024

Reviewers' comments:

Reviewer's Responses to Questions

Comments to the Author

1. Is the manuscript technically sound, and do the data support the conclusions?

Reviewer #1: Partly

Reviewer #2: Yes

Response: We revised the whole text and conclusion carefully using red markers. 

2. Has the statistical analysis been performed appropriately and rigorously?

Reviewer #1: Yes

Reviewer #2: Yes

3. Have the authors made all data underlying the findings in their manuscript fully available?

Reviewer #1: Yes

Reviewer #2: Yes

4. Is the manuscript presented in an intelligible fashion and written in standard English?

Reviewer #1: No

Reviewer #2: Yes

Response: We revised the English of the text carefully using red markers by American Journal Experts (AJE). We believe that the revised manuscript is more readable.

5. Review Comments to the Author

Reviewer #1: The study of Wu et al. domenstrates the application of chloroplast genomes in explore the germplasm of Bougainvillea cultivars. Apart from the phylogenetic relationships between the cultivars, potential molecular markers including SSRs, LSRs and hotspot regions have been investigated from the chloroplast genomes. The study shows merits on the research of Bougainvillea, that could serve as a reference in studying other horticultural crops with diversified morphologies. However, prior to a make a further decision, the following major and minor issues should be resolved.

Major issues:

1) The cultivars of Bougainvillea are regulated by the International Code for the Nomenclature for Cultivated Plants (ICNCP). The registration of Bougainvillea cultivars is designated to the Indian Agricultural Research Institute (IARI). The authors are responsible to carefully check if the cultivar epithets are well established and registered. Illegitimate epithets (e.g. misspelling and homonyms) and unestablished epithets are common in horticultural germplasms. The authors should refer to the Articles 25 to 27 of ICNCP (9th Edition) which is available online (https://www.ishs.org/sites/default/files/static/ScriptaHorticulturae_18.pdf). The publications of The Bougainvillea Society of India (BSI) of IARI (http://www.bsi-iari.com/publication.htm) will help the authors in checking. The authors should indicate those unestablished and unregistered cultivars in the manuscript as precaution for the readers.

Response: Thank you for this question. We can’t open the BSI of IARI. So the cultivars in the manuscript were unregistered in this web. We wrote in Data Availability Statement as following: The 13 Bougainvillea cultivars studied in this study are unregistered in the Bougainvillea Society of India (BSI) of IARI (http://www.bsi-iari.com/publication.htm). 

2) I wonder if the mutant (Bougainvillea sp.1) is discovered by the authors themselves? 

Response: Thank you for this question. Yes, the mutant (Bougainvillea sp.1) was discovered by the author He-Fa Wang. 

If so, what is the original cultivar of this mutant? 

Response: The mutant (Bougainvillea sp.1) was from the other cultivar (mother line), of which the bract colour looked like the bract colour of the cultivar Bougainvillea ‘Barbara Karst’. This mother line has thorns and was sold out in commercial activities. However, the bud mutant was kept in live and grown by grafting. Because the author He-Fa Wang hasn’t yet got certificate of registered commercial name for the mutant. In the manuscript, we gave the mutant with the name of Bougainvillea sp.1. 

The authors should elaborate why this mutant is included in this study.

Response: In this study, we want to know the phylogenetic relationships among the mutant (Bougainvillea sp.1) and other individuals of the Bougainvillea genus by using complete chloroplast genomes. 

 Also, Figure S1 (the images of 13 cultivars) should be included in the main text, prior to the chloroplast genome map. Vouchers of the studied cultivars, either live and dried specimens, should be listed out with their collection location, GPS, date of collection, collector numbers, and deposited herbarium or institution.

Response: Thank you for this question. We revised Figure 1 instead Figure S1 in the manuscript. In the Plant materials, DNA extraction, and sequencing section, we wrote detail information of the studied cultivars, either live and dried specimens, with their collection location, GPS, date of collection, collector numbers, and deposited herbarium or institution.

Fresh leaves of twelve Bougainvillea cultivars and one bud mutation, namely, B.× buttiana ‘Mahara’ , B. × buttiana ‘Gautama's Red’, B. × buttiana ‘California Gold’, B. × buttiana ‘Double Salmon’, B. × buttiana ‘Double Yellow’, B. × buttiana ‘Big Chitra’, B. × buttiana ‘Los Banos Beauty’, B. glabra ‘White Stripe’, B. spectabilis ‘Flame’, B. spectabilis ‘Splendens’, B. ‘Barbara Karst’, B. ‘San Diego Red’, and B. sp. 1 (Fig 1, S1 Table), were collected from the Provincial Flower Germplasm Resources Bank of San Jiao Mei (117°37′47″E, 24°28′35″N) in Zhangzhou, Fujian Province, China. Fresh leaves were quickly frozen on dry ice, sent to the laboratory of the Environmental Horticulture Research Institute at the Guangdong Academy of Agricultural Sciences (113°21′8″E, 23°9′2″N), Guangzhou, China, and stored at −80 ℃ until use. Genomic chloroplast DNA was extracted from each sample using the modified sucrose gradient centrifugation method [13]. Then, the DNA quality and quantity were checked through agarose gel electrophoresis and the NanoDrop microspectrometer method, respectively. Each qualified DNA sample was sheared to fragments of approximately 350 bp. Short-insert (350 bp) paired-end libraries were constructed, and sequencing was performed on an Illumina NovaSeq 6000 platform with a paired read length of 150 bp (Biozeron, Shanghai, China). The raw data from each sample were checked using FastQC v. 0.11.9 (http://www.bioinformatics.babraham.ac.uk/projects/fastqc/), and adaptors and low-quality reads were subsequently deleted by Trimmomatic v. 0.39 [14] with default parameters. The remaining materials, including the leaves and DNA, were deposited in the laboratory of the Environmental Horticulture Research Institute (store sheet code: B2023), Guangdong Academy of Agricultural Sciences, Guangzhou, China, as vouchers (S1 Table).

Fig 1. Morphologies among 13 cultivars of the Bougainvillea genus. A, Bougainvillea glabra ‘White Stripe’; B, Bougainvillea×buttiana ‘Mahara’; C, Bougainvillea×buttiana ‘Double Yellow’; D, Bougainvillea×buttiana ‘Double Salmon’; E, Bougainvillea×buttiana ‘Los Banos Beauty’; F, Bougainvillea ×buttiana ‘California Gold’; G, Bougainvillea ×buttiana ‘Big Chitra’; H, Bougainvillea×buttiana ‘Gautama's Red’; I, Bougainvillea spectabilis ‘Flame’; J, Bougainvillea spectabilis ‘Splendens’; K, Bougainvillea ‘Barbara Karst’; L, Bougainvillea ‘San Diego Red’; M and N, Bougainvillea sp1. armed with simple or no thorns. 

3) I noticed that Bougainvillea spinosa was included in Clade II by the authors. However, it should not be regarded as a member of Clade II, as it is sister to all other members from both clade II and III (in both Figure 8 A&B and S3 A&B). The authors should discuss the potential reasons in the Discussion.

Response: Thank you for this question. Bougainvillea spinosa should be clustered in Clade Ⅰ. We revised the results of phylogenetic relationships as following.

In Clade Ⅰ, B. spectabilis ‘Flame’ was sister to B. peruviana MW123901, then forming a cluster strongly sister to B. pachyphylla, both based on chloroplast genome sequences and protein-coding genes (Fig 9, S2 Fig). However, the position of B. × buttiana ‘Gautama's Red’ in ML tree constructed by chloroplast genomes sequences was different from the other three phylogenetic trees in this study. For the former, B. × buttiana ‘Gautama's Red’ was sister to B. spectabilis ‘Pixie Pink’, then clustered with B. peruviana MT407463 with strong support (BS = 90%) (Fig 9A). For the latter, B. spectabilis ‘Pixie Pink’ was sister to B. peruviana MT407463, then clustered with B. × buttiana ‘Gautama's Red’ with strong support (BS = 93-100%, and PP = 0.93-1) (Fig 9B, S2 Fig). Then, B. spinosa and these 6 individuals were clustered together with strong support in Clade Ⅰ (Fig 9, S2 Fig). In Clade Ⅱ, there were 7 individuals of wild species, including B. campanulata, B. berberidifolia, B. infesta, B. modesta OM44398, B. modesta OM044396, B. stipitata, and B. stipitata var. grisebachiana.

4)The definition of SNPs and InDels in this manuscript should be well defined. Could the nucleotide differences between one and another cultivar be considered as SNPs and InDels? 

Molecular Diagnostic Characters (MDCs) of a cultivar to differentiate itself from the other 12 cultivars could be more meaningful. Also, haplotype analysis could aid in visualizing the figures in this tables by grouping the cultivars. The authors could refer to the following article:

Wong, K. H., Wu, H. Y., Kong, B. L. H., But, G. W. C., Siu, T. Y., Hui, J. H. L., Shaw, P. C., & Lau, D. T. W. (2022). Characterisation of the complete chloroplast genomes of seven Hyacinthus orientalis L. cultivars: Insights into cultivar phylogeny. Horticulturae, 8(5), 1-28.

Response: Thank you for this question. We didn’t use the presentation of SNP and indels as described in cultivars of Hyacinthus orientalis. For one reason, in my opinion, the background of Hyacinthus orientalis was less complicated than the background of the cultivars of Bougainvillea. The breeding of Hyacinthus orientalis didn’t use grafting; however, grafting is very common in cultivation of Bougainvillea cultivars. For other reason, too many SNPs and indels (more than 100), such as B. glabra ‘White Stripe’ vs. B. × buttiana ‘Gautama's Red’, B. glabra ‘White Stripe’ vs. B. spectabilis ‘Flame’ and B. × buttiana ‘Mahara’ vs. B. × buttiana ‘Gautama's Red’, were not suitable in the text. We also showed the SNPs and indels of different comparisons in the supplementary file (S7 Table). 

Minor issues:

The writing style and language of the manuscript, particularly the abstracts, introduction and the discussion, should be meticulously improved. I have identified a number of mistakes in grammar and use of wordings which deteriorate the comprehensibility of the manuscript. In addition, I highly recommend the authors to employ a native English speaker or professional editing agency to proofread the manuscript. The style in presenting figures and tables should also be improved. Please kindly refer to the following:

L16-18: Having high similarity in leaf appearance and hybridization among Bougainvillea species, the phylogenetic relationships of the genus are complicated and controversial.

Response: Thank you for this idea. We revised as following:

Having high similarity in leaf appearance and hybridization among Bougainvillea species, especially from Bougainvillea × buttiana, their phylogenetic relationships were very complicated and controversial. 

L20-21: Their phylogenetic relationships within the genus Bougainvillea and other species of the family Nyctaginaceae are identified for the first time.

Response: Thank you for this idea. We revised the sentence as following:

In this study, we sequenced, assembled and analyzed thirteen complete chloroplast genomes of Bougainvillea cultivars from South China, including seven B. × buttiana cultivars and six other Bougainvillea cultivars, and identified their phylogenetic relationships within the Bougainvillea genus and other species of the Nyctaginaceae family for the first time. 

L27-28: Four divergent regions, including ......, were identified from sliding window analysis of 16 Bougainvillea cultivar genomes.

Response: Thank you for this idea. Comparative analysis not only contained sliding window analysis (mVISTA), but also contained CGView analysis and nuclear diversity analysis (Pi). Therefore, we revised the sentence as following:

Four divergent regions, including trnH-GUG_psbA, trnS-GCU_trnG-UCC-exon1, trnS-GGA_rps4, and ccsA_ndhD, were identified from comparative analysis of 16 Bougainvillea cultivars genomes. 

L36: replace "which contained" by "including"

Response: Thank you for this idea. We revised ‘including’ instead of ‘which contained’.

L39: ..., but also helped to identify Bougainvillea

Response: Thank you for this idea. We revised ‘helped to identify Bougainvillea’

L51: "Le Du Juan", which is well-known in China, belongs to the genus Bougainvillea of the family Nyctaginaceae.

Response: Thank you for this idea. We revised as following:

 “Le Du Juan”, which is well-known in China, belongs to the Bougainvillea genus of the Nyctaginaceae family. 

L53-54: with colored bracts [2,3]. The colorful bracts surrounding the small tubular flowers are often mistakenly treated as flowers.

Response: Thank you for this idea. We revised as following:

The colorful bracts surrounding the small tubular flowers are often mistakenly treated as flowers.

L59: remove "value"

Response: Thank you for this idea. We removed ‘value’.

L60: replace "only" by "mainly"

Response: Thank you for this idea. We revised ‘mainly’ instead of ‘only’.

L60-61: ... challenging because of high similarity

Response: Thank you for this idea. We revised ‘challenging because of high similarity’. 

L68: replace "identified by" by "explored using"

Response: Thank you for this idea. We revised ‘explored using’ instead of ‘identified by’. 

L69: Why is this sentence concerning the usage of Bougainvillea is placed here? Recommend placing it in the previous paragraphs.

Response: Thank you for this idea. We placed this sentence in the previous paragraph.

L70: delete "that has been"

Response: Thank you for this idea. We deleted ‘that has been’. 

L71: replace "taken" by "introduce"

Response: Thank you for this idea. We revised ‘introduce’ instead of ‘taken’. 

L75: ... B. x buttiana cultivars and the molecular evolution ...

Response: Thank you for this idea. We revised as following: 

However, phylogenetic relationships of B. × buttiana cultivars and molecular evolution of chloroplast genomes from the Nyctaginaceae family still remains to be unveiled [2-5,9-12]. 

L77-78: In this study, complete chloroplast genomes of thirteen Bougaivillea cultivars were newly sequence, assembled and annotated. These thirteen cultivars from South China include seven ....

Response: Thank you for this idea. We revised as following: 

In this study, complete chloroplast genomes of thirteen Bougainvillea cultivars were newly sequenced, assembled and annotated. These thirteen cultivars from South China included seven B. × buttiana cultivars, 

L84: ... integrating three published ...

Response: Thank you for this idea. We revised as following: 

 by integrating three published complete chloroplast genomes 

---

## [Decision Letter · Decision Letter 1]

7 May 2024

PONE-D-23-43492R1Comparative analysis of the complete chloroplast genomes of thirteen Bougainvillea cultivars from South China with implications for their genome structures and phylogenetic relationshipsPLOS ONE

Dear Dr. Li,

Thank you for submitting your manuscript. Following a detailed review, our editorial team and reviewers have identified several critical areas that require revision before we can reconsider your manuscript for publication. We believe that your work has potential, but significant changes are needed to ensure the quality and accuracy of your research.

Please review the comments from our reviewers below, and provide a detailed response letter outlining how you have addressed each point. The revised manuscript should reflect these changes and adhere to our journal's guidelines.

Key Issues for Revision:

One of the reviewers noted inconsistencies in the cultivar nomenclature. You must confirm the legitimacy of the names used and correct any discrepancies. Additionally, the clustering of Bougainvillea spinosa into clades appeared inconsistent, suggesting a need for further consultation with experts in phylogeny or re-analysis.

Another reviewer highlighted the absence of references to existing studies on the chloroplast genome of Bougainvillea. Please ensure you incorporate relevant literature, including a comparative analysis with studies like Lin et al. (2023). Discuss any differences in findings and explain their significance.

The reviewers requested more context regarding the significance of the thornless mutant. Additionally, they suggested examining intraspecific genetic distances among cultivars derived from the same species, particularly B. x buttiana and B. spectabilis.

It was noted that you constructed multiple phylogenetic trees, but the differences between Maximum Likelihood and Bayesian Inference were not explained. Please clarify why these methods were used and which types of sequences are most suitable for constructing phylogenetic trees.

Reviewers identified additional minor issues, such as the correct presentation of GPS coordinates, language and format consistency, and a more reader-friendly presentation of SNPs and Indels. Address these issues to improve the clarity and readability of the manuscript.

Given these concerns, we request that you revise your manuscript and submit a detailed response letter that explains how you have addressed each point. Please submit your revised manuscript by [Deadline Date]. If you need more time, do let us know, and we will be happy to discuss an extension.

We look forward to receiving your revised manuscript. If you have any questions or need further guidance, please do not hesitate to contact us.

Thank you for your understanding and cooperation.

Sincerely,

We look forward to receiving your revised manuscript.

Kind regards,

Pankaj Bhardwaj, Ph.D.

Academic Editor

PLOS ONE

Reviewers' comments:

Reviewer's Responses to Questions

**Comments to the Author**

1. If the authors have adequately addressed your comments raised in a previous round of review and you feel that this manuscript is now acceptable for publication, you may indicate that here to bypass the “Comments to the Author” section, enter your conflict of interest statement in the “Confidential to Editor” section, and submit your "Accept" recommendation.

Reviewer #1: (No Response)

Reviewer #3: (No Response)

Reviewer #4: (No Response)

2. Is the manuscript technically sound, and do the data support the conclusions?

Reviewer #1: Partly

Reviewer #3: Yes

Reviewer #4: Yes

3. Has the statistical analysis been performed appropriately and rigorously? 

Reviewer #1: Yes

Reviewer #3: Yes

Reviewer #4: Yes

4. Have the authors made all data underlying the findings in their manuscript fully available?

Reviewer #1: Yes

Reviewer #3: Yes

Reviewer #4: Yes

5. Is the manuscript presented in an intelligible fashion and written in standard English?

Reviewer #1: Yes

Reviewer #3: Yes

Reviewer #4: Yes

6. Review Comments to the Author

Reviewer #1: It is highly appreciated that the authors have made substantial efforts in addressing most of the minor issues and improving comprehensibility of the manuscript. However, the major issues were not settled, particularly the cultivar nomenclature and the clustering of clades. It is extremely irresponsible that the authors calimed all the cultivars unregistered only because of their inaccessiblity to the website of Bougainvillea Society of India. Some of the studied cultivars were registered while some are not. This act is seriously misleading to the readers since they claimed some legitimate cultivar epithets as illegitimate. The authors should try their best to reach to the pieces of information regardless of any format (e.g. books, printed literatures and any electronic means), but not just giving up and leave an incorrect and irrespinsible statement. For the clustering of Bougainvillea spinosa, the authors obviously showed their wrong concept on monophyly. The claims of the authors (Bougainvillea spinosa was in either clade I or II) is very misleading to the readers outside the field of phylogeny.

The authors clearly failed to address all four major issues. For major #2, the significancy in including the thornless mutant was not well explained and elaborated. For major #4, the breeding techniques do not affect the presentation of SNPs and Indels which are significant in cultivar authentication. It is understandable their are plenty of SNPs discovered from the analysis. Yet, the authors failed the listed out those valuable in cultivar authentication in an reader-friendly manner. For major #1 and #3, the authors should obtain more suggestions froms experienced taxonomists and phylogenists, respectively. A numbers of minor issues were also not well addressed. The authors should read more articles to find out which format "the xxx genus/family" or "the genus/family xxx" should be correct in English. The authors also failed to add authorities after genus name and species epithets, which are commonly adopted in the scientific research articles. The presentation of GPS coordinates was in wrong order. Suggestions on figure and table presentation were sadly ignored.

This interesting sutdy in exploring cultivar phylogeny of Bougainvillea would potentially contribute to the science community. Although the authors have made plenties of efforts in improving the manuscript, it is sorrow that I have to suggest a rejection from publishing this articles delivering incorrect information.

Reviewer #3: The overall write up is convincing. As this work is on the characterization of Bougainvillea cultivars, it should not be related to anything about taxonomy. Eventually, I have two concerns for this manuscripts:

1. There are a number of published work on the complete chloroplast genome of Bougainvillea. Considering that all information are important, I would be looking at a discussion on the difference of findings between this work and other published works, and the reasons for the difference identified in these analyses. For example, Lin et al. 2023, International Journal of Molecular Science 24:15138 should be the latest work before this work. I neither see it cited, nor being discussed. Lin et al. had similar analyses when compared to the current one. Please make sure all published work on the complete chloroplast genome of Bougainvillea are included in this work and compared for their findings.

2. Since some of the cultivars are named along with their parent (perhaps they are mutant clones), it would be reasonable to at least investigate the intraspecific genetic distance of these cultivar derived from the (presumably) same species. At least the analysis is conducted on the species assembled in this study, i.e., B. x buttiana and B. spectabilis.

Reviewer #4: The manuscript titled 'Comparative analysis of the complete chloroplast genomes of thirteen Bougainvillea cultivars from South China with implications for their genome structures and phylogenetic relationships' presents the assembly and annotation of 13 cp genomes of Bougainvillea. The study is well-organized and offers valuable insights for analyzing genetic diversity and phylogenetic relationships within the Nyctaginaceae family. I highly recommend this work for publication in the PLoS One journal.

I think the following minor suggestions and comments will help the authors improve their work.

1. The authors should provide additional information of the origin of the bud mutation, such as which plant accession did this mutation derive from?

2. I wondered why these authors constructed so many types of phylogenetic trees, including two ML and BI trees based on whole cp genome sequences and coding sequences. These authors should point out what’s the difference between ML and BI trees, and which sequence is more appropriate to construct phylogenetic tree.

7. PLOS authors have the option to publish the peer review history of their article (what does this mean?). If published, this will include your full peer review and any attached files.

Reviewer #1: No

Reviewer #3: No

Reviewer #4: No

---

## [Author Response · Author response to Decision Letter 1]

11 Jun 2024

Reviewer #1: It is highly appreciated that the authors have made substantial efforts in addressing most of the minor issues and improving comprehensibility of the manuscript. However, the major issues were not settled, particularly the cultivar nomenclature and the clustering of clades. It is extremely irresponsible that the authors claimed all the cultivars unregistered only because of their inaccessiblity to the website of Bougainvillea Society of India. Some of the studied cultivars were registered while some are not. This act is seriously misleading to the readers since they claimed some legitimate cultivar epithets as illegitimate. The authors should try their best to reach to the pieces of information regardless of any format (e.g. books, printed literatures and any electronic means), but not just giving up and leave an incorrect and irrespinsible statement. 

Response: We removed the statement. The 12 Bougainvillea cultivars included B. × buttiana ‘Mahara’ , B. × buttiana ‘California Gold’, B. × buttiana ‘Double Yellow’, B. × buttiana ‘Los Banos Beauty’, B. × buttiana ‘San Diego Red’, B. ×buttiana ‘Barbara Karst’, B. spectabilis ‘Flame’, B. spectabilis ‘Splendens’, B. × buttiana ‘Gautama's Red’, B. × buttiana ‘Double Salmon’, B. × buttiana ‘Big Chitra’, and B. glabra ‘White Stripe’. Their names came from following information: 

Liu Y, Ruan L, Zhou H, Yu M. Cultivar classification of Bougainvillea. China Forestry Press, Beijing, China, 2020

Sun L. Molecular identification of cultivars and transcriptome analysis of bracts in Bougainvillea. Ph. D. Chinese Academy of Forestry, Beijing, China, 2019, pp38-39

From the description of Sun in 2019, B.× buttiana ‘San Diego Red’ and B. ×buttiana ‘Barbara Karst’ were the correct names for Bougainvillea ‘San Diego Red’ and Bougainvillea ‘Barbara Karst’, respectively. We used B. × buttiana ‘San Diego Red’ and B. ×buttiana ‘Barbara Karst’ in the whole text instead of B. ‘San Diego Red’ and B. ‘Barbara Karst’, respectively. 

Because the thornless mutant was a variety from Bougainvillea × buttiana ‘Miss Manila’ and the author He-Fa Wang hasn’t yet got certificate of registered commercial name for the mutant, we gave the mutant with the name of Bougainvillea × buttiana ‘Miss Manila’ sp.1 in the text. 

For the clustering of Bougainvillea spinosa, the authors obviously showed their wrong concept on monophyly. The claims of the authors (Bougainvillea spinosa was in either clade I or II) is very misleading to the readers outside the field of phylogeny.

Response: Thanks for your opinion on the clustering of Bougainvillea spinosa. 

 In Bautista et al. (2022), B. spinosa was sister to clade II or wild clade Bougainvillea in that study. However, Lin et al. (2023) clustered the B. spinosa into clade 3, which was the same clade in our study, clade I. 

Bautista MAC, Zheng Y, Boufford DE, Hu Z, Deng Y, Chen T. Phylogeny and taxonomic synopsis of the genus Bougainvillea (Nyctaginaceae). Plants (Basel). 2022; 11, 1700. https://doi.org/10.3390/plants11131700 PMID: 35807654

Lin X, Lee SY, Ni J, Zhang X, Hu X, Zou P, Wang W, Liu G. Comparative analyses of chloroplast genome provide effective molecular markers for species and cultivar identification in Bougainvillea. Int J Mol Sci. 2023; 24(20),15138. https://doi.org/10.3390/ijms242015138. PMID: 37894819

We revised the section Phylogenetic relationships in the Bougainvillea genus as following:

The 35 Bougainvillea individuals analyzed were divided into four clades, namely, Clades Ⅰ, Ⅱ, Ⅲ, and Ⅳ, with strongly supported values (BS = 85–100% for the ML trees and PP = 0.99–1 for the BI trees) (Fig 9, S2 Fig). Two cultivars, B. × buttiana ‘Gautama's Red’ and B. spectabilis ‘Flame’, were clustered into clade Ⅰ, and the other 11 cultivars, including B. × buttiana ‘Mahara’, B. × buttiana ‘California Gold’, B. × buttiana ‘Double Salmon’, B. × buttiana ‘Double Yellow’, B. × buttiana ‘Los Banos Beauty’, B. × buttiana ‘Big Chitra’, B. × buttiana ‘Barbara Karst’, B. glabra ‘White Stripe’, B. spectabilis ‘Splendens’, B. × buttiana ‘San Diego Red’, and B. × buttiana ‘Miss Manila’ sp. 1, were clustered into clade Ⅳ (Fig 9, S2 Fig). In Clade Ⅰ, B. spectabilis ‘Flame’ was sister to B. peruviana MW123901 and then formed a strong sister cluster to B. pachyphylla, both based on chloroplast genome sequences and protein-coding genes (Fig 9, S2 Fig). However, the position of B. × buttiana ‘Gautama's Red’ in the ML tree constructed from chloroplast genome sequences differed from those in the other three phylogenetic trees in this study. For the former, B. × buttiana ‘Gautama's Red’ was sister to B. spectabilis ‘Pixie Pink’ and then clustered with B. peruviana MT407463 with strong support (BS = 90%) (Fig 9A). For the latter, B. spectabilis ‘Pixie Pink’ was sister to B. peruviana MT407463 and then clustered with B. × buttiana ‘Gautama's Red’ with strong support (BS = 93–100%, and PP = 0.93–1) (Fig 9B, S2 Fig). In Clade Ⅱ, it only contained B. spinosa (Fig 9, S2 Fig). In Clade Ⅲ, there were 7 individuals of wild species, including B. campanulata, B. berberidifolia, B. infesta, B. modesta OM44398, B. modesta OM044396, B. stipitata, and B. stipitata var. grisebachiana. In Clade Ⅳ, in the ML tree based on the chloroplast genome sequences, B. × buttiana ‘Mahara’, B. × buttiana ‘California Gold’, B. × buttiana ‘Double Salmon’, B. × buttiana ‘Double Yellow’, B. × buttiana ‘Los Banos Beauty’, B. × buttiana ‘Big Chitra’, B. × buttiana ‘San Diego Red’, B. spectabilis ‘Ratana Red’, B. glabra MN888961, and B. peruviana ‘Mona Lisa Red’ were clustered together in one cluster with strong support (BS = 88–92%). B. × buttiana ‘Barbara Karst’, B. glabra ‘White Stripe’, B. spectabilis ‘Splendens’, B. × buttiana ‘Miss Manila’ sp. 1, B. spectabilis MN315508, B. spectabilis China MW167297, and B. hybrid cultivar MW123903 were clustered together in another cluster with strong support (BS =88–95%) (Fig 9A). However, in the ML tree based on protein-coding genes, B. × buttiana ‘Mahara’ was sister to the other cultivars in Clade Ⅳ with strong support (BS = 100%) (S2A Fig). In both BI trees, B. × buttiana ‘Mahara’, B. × buttiana ‘California Gold’, B. × buttiana ‘Double Salmon’, B. × buttiana ‘Double Yellow’, B. × buttiana ‘Los Banos Beauty’, B. × buttiana ‘Big Chitra’, B. × buttiana ‘San Diego Red’, B. spectabilis ‘Ratana Red’, B. glabra, B. peruviana ‘Mona Lisa Red’, B. × buttiana ‘Barbara Karst’, B. glabra ‘White Stripe’, B. spectabilis ‘Splendens’, B. × buttiana ‘Miss Manila’ sp. 1, B. spectabilis MN315508, B. spectabilis MW167297, and B. hybrid cultivar MW123903 were clustered together in one cluster in Clade Ⅳ with moderate to strong support (PP = 0.84–1) (Fig 9B, S2B Fig). In the four phylogenetic trees, Clades Ⅲ and Ⅳ were clustered together, forming a cluster with strong support (BS =85–99%, and PP = 0.99–1); then the cluster, Clade Ⅱ, and Clade Ⅰ were clustered step by step in the Bougainvillea genus with strong support (BS = 99-100%, and PP = 0.99-1) (Fig 9, S2 Fig). 

The authors clearly failed to address all four major issues. For major #2, the significancy in including the thornless mutant was not well explained and elaborated.

Response: Thanks for your idea. The thornless mutant was a variety from Bougainvillea × buttiana ‘Miss Manila’. The mother line of Bougainvillea × buttiana ‘Miss Manila’ has been sold out in commercial activities so far. However, the bud mutant has been kept in live and cultivated by grafting. Because the author He-Fa Wang hasn’t yet got certificate of registered commercial name for the mutant. In the manuscript, we gave the mutant with the name of Bougainvillea × buttiana ‘Miss Manila’ sp.1. 

For major #4, the breeding techniques do not affect the presentation of SNPs and Indels which are significant in cultivar authentication. It is understandable their are plenty of SNPs discovered from the analysis. Yet, the authors failed the listed out those valuable in cultivar authentication in an reader-friendly manner. 

Response: Thanks for your idea. We didn’t use the presentation of SNP and indels as described in cultivars of Hyacinthus orientalis. For three comparisons, there were more than 700 SNPs and 100 indels. This situation is not fit in the text. However, we listed out these SNPs and indels in detail information in the supplementary file: S7 Table, including base information of SNPs, mutant type, gene name, gene start, gene end, indel sequence, indel start, indel end, alignment strand and so on. 

For major #1 and #3, the authors should obtain more suggestions froms experienced taxonomists and phylogenists, respectively. A numbers of minor issues were also not well addressed. The authors should read more articles to find out which format "the xxx genus/family" or "the genus/family xxx" should be correct in English. 

Response: Thanks for your idea. These two forms have appeared in many articles. We selected the form of “the XXX genus/family” in the text. 

The authors also failed to add authorities after genus name and species epithets, which are commonly adopted in the scientific research articles. 

Response: The name of 12 Bougainvillea cultivar were adopted in the scientific book as following.

Liu Y, Ruan L, Zhou H, Yu M. Cultivar classification of Bougainvillea. China Forestry Press, Beijing, China, 2020

Sun L. Molecular identification of cultivars and transcriptome analysis of bracts in Bougainvillea. Ph. D. Chinese Academy of Forestry, Beijing, China, 2019, pp38-39

The presentation of GPS coordinates was in wrong order. 

Response: We checked and revised the correct GPS in correct order. We reported details about the GPS for sample collection, DNA sample store and the rest leaf materials store.

Fresh leaves of twelve Bougainvillea cultivars, namely, B.× buttiana ‘Mahara’ , B. × buttiana ‘Gautama's Red’, B. × buttiana ‘California Gold’, B. × buttiana ‘Double Salmon’, B. × buttiana ‘Double Yellow’, B. × buttiana ‘Big Chitra’, B. × buttiana ‘Los Banos Beauty’, B. glabra ‘White Stripe’, B. spectabilis ‘Flame’, B. spectabilis ‘Splendens’, B. × buttiana ‘Barbara Karst’, and B. × buttiana ‘San Diego Red’ (Fig 1, S1 Table), were collected from the Provincial Flower Germplasm Resources Bank of San Jiao Mei in Zhangzhou (117°37′47″E, 24°28′35″N), Fujian Province, China. One bud mutation armed with simple or no thorns and derived from B. × buttiana ‘Miss Manila’, given namely, B. × buttiana ‘Miss Manila’ sp. 1 (Fig 1, S1 Table), was collected from the cultivation factoty of Zhangzhou (117°49′9″E, 24°31′33″N) in Xiamen Qianrihong Horticulture Co., Ltd, Fujian Province, China. Fresh leaves were quickly frozen on dry ice, sent to the laboratory of the Environmental Horticulture Research Institute (113°20′39″E, 23°8′51″N) at the Guangdong Academy of Agricultural Sciences, Guangzhou, China, and stored at −80 ℃ until use. Genomic chloroplast DNA was extracted from each sample using the modified sucrose gradient centrifugation method [13]. Then, the DNA quality and quantity were checked through agarose gel electrophoresis and the NanoDrop microspectrometer method, respectively. Each qualified DNA sample was sheared to fragments of approximately 350 bp. Short-insert (350 bp) paired-end libraries were constructed, and sequencing was performed on an Illumina NovaSeq 6000 platform with a paired read length of 150 bp (Biozeron, Shanghai, China). The raw data from each sample were checked using FastQC v. 0.11.9 (http://www.bioinformatics.babraham.ac.uk/projects/fastqc/), and adaptors and low-quality reads were subsequently deleted by Trimmomatic v. 0.39 [14] with default parameters. The remaining materials, including the leaves and DNA, were deposited in the laboratory of the Environmental Horticulture Research Institute (store sheet code: B2023, 113°20′39″E, 23°8′51″N), Guangdong Academy of Agricultural Sciences, Guangzhou, China, as vouchers (S1 Table).

Suggestions on figure and table presentation were sadly ignored.

Response: We revised Figure 1 to Figure 9 as following. 

We did not ignore the presentation of Table 4. For table 4, ‘462 L 0.969’ and ‘462 |L|0.969’ were two different forms for presentation. We use the form of ‘462 L 0.969’, not used the form of ‘462 |L|0.969’. Because the analyzed result from the CodeML program used the form of ‘462 L 0.969’, not used the form of ‘462 |L|0.969’. 

Reviewer #3: The overall write up is convincing. As this work is on the characterization of Bougainvillea cultivars, it should not be related to anything about taxonomy. Eventually, I have two concerns for this manuscripts:

1. There are a number of published work on the complete chloroplast genome of Bougainvillea. Considering that all information are important, I would be looking at a discussion on the difference of findings between this work and other published works, and the reasons for the difference identified in these analyses. For example, Lin et al. 2023, International Journal of Molecular Science 24:15138 should be the latest work before this work. I neither see it cited, nor being discussed. Lin et al. had similar analyses when compared to the current one. Please make sure all published work on the complete chloroplast genome of Bougainvillea are included in this work and compared for their findings.

Response: Thank you for this idea. We cited the article (Lin et al. 2023), compared and discussed the chloroplast genome, phylogenetic tree and related results. 

Lin X, Lee SY, Ni J, Zhang X, Hu X, Zou P, Wang W, Liu G. Comparative analyses of chloroplast genome provide effective molecular markers for species and cultivar identification in Bougainvillea. Int J Mol Sci. 2023; 24(20),15138. https://doi.org/10.3390/ijms242015138. PMID: 37894819

2.Since some of the cultivars are named along with their parent (perhaps they are mutant clones), it would be reasonable to at least investigate the intraspecific genetic distance of these cultivar derived from the (presumably) same species. At least the analysis is conducted on the species assembled in this study, i.e., B. x buttiana and B. spectabilis.

Response: This is a good question. There were two main reasons that we did not sequence and assemble the wild species of B.×buttiana and B. spectabilis. First, wild species of B. spectabilis had been sequenced and reported in previous studies before, such as in Wang et al. 2019 and Bautista et al. 2020, 2022. Second, in the Provincial Flower Germplasm Resources Bank of San Jiao Mei in Zhangzhou (117°37′47″E, 24°28′35″N), Fujian Province, China, there are so many cultivars of B.×buttiana. However, the wild species of B.×buttiana was not easy to be obtained and identified. Therefore, we used the published chloroplast genomes of B. spectabilis in NCBI for phylogenetic analysis. 

Based on this suggestion, we added the SNPs/indels analyses among groups of B.×buttiana ‘Mahara’ vs. B. spectabilis ‘Flame’, B.×buttiana ‘Mahara’ vs. B. spectabilis ‘Splendens’, and B. spectabilis ‘Splendens’-OR253994 (Lin et al. 2023) vs. B. spectabilis ‘Splendens’-OR344372 (this study). The details results can be found in two sections of SNPs and indels analyses among the thirteen complete chloroplast genomes, and Intraspecific analyses of two chloroplast genomes of B. spectabilis ‘Splendens’. 

Bautista MAC, Zheng Y, Boufford DE, Hu Z, Deng Y, Chen T. Phylogeny and taxonomic synopsis of the genus Bougainvillea (Nyctaginaceae). Plants (Basel). 2022; 11, 1700. https://doi.org/10.3390/plants11131700 PMID: 35807654

Bautista MAC, Zheng Y, Hu Z, Deng Y, Chen T. Comparative analysis of complete chloroplast genome sequences of wild and cultivated Bougainvillea (Nyctaginaceae). Plants (Basel). 2020; 9, 1671. https://doi.org/10.3390/plants9121671 PMID: 33260641

Lin X, Lee SY, Ni J, Zhang X, Hu X, Zou P, Wang W, Liu G. Comparative analyses of chloroplast genome provide effective molecular markers for species and cultivar identification in Bougainvillea. Int J Mol Sci. 2023; 24(20),15138. https://doi.org/10.3390/ijms242015138. PMID: 37894819

Wang N, Qiu MY, Yang Y, Li JW, Zou XX. Complete chloroplast genome sequence of Bougainvillea spectabilis (Nyctaginaceae). Mitochondrial DNA B Resour. 2019; 4, 4010-4011. https://doi.org/ 10.1080/23802359.2019.1688716 PMID: 33366293

Reviewer #4: The manuscript titled 'Comparative analysis

---

## [Decision Letter · Decision Letter 2]

26 Aug 2024

Comparative analysis of the complete chloroplast genomes of thirteen Bougainvillea cultivars from South China with implications for their genome structures and phylogenetic relationships

PONE-D-23-43492R2

Dear Dr. Li,

We’re pleased to inform you that your manuscript has been judged scientifically suitable for publication and will be formally accepted for publication once it meets all outstanding technical requirements.

Kind regards,

Pankaj Bhardwaj, Ph.D.

Academic Editor

PLOS ONE

Additional Editor Comments (optional):

Reviewers' comments:

Reviewer's Responses to Questions

**Comments to the Author**

1. If the authors have adequately addressed your comments raised in a previous round of review and you feel that this manuscript is now acceptable for publication, you may indicate that here to bypass the “Comments to the Author” section, enter your conflict of interest statement in the “Confidential to Editor” section, and submit your "Accept" recommendation.

Reviewer #4: All comments have been addressed

Reviewer #5: All comments have been addressed

2. Is the manuscript technically sound, and do the data support the conclusions?

Reviewer #4: Yes

Reviewer #5: Yes

3. Has the statistical analysis been performed appropriately and rigorously? 

Reviewer #4: Yes

Reviewer #5: Yes

4. Have the authors made all data underlying the findings in their manuscript fully available?

Reviewer #4: Yes

Reviewer #5: Yes

5. Is the manuscript presented in an intelligible fashion and written in standard English?

Reviewer #4: Yes

Reviewer #5: Yes

6. Review Comments to the Author

Reviewer #4: (No Response)

Reviewer #5: After reading the revised MS, they have answered all raised questions. the plastid genome and all kinds of methods to build the phylogeny is solid for this paper.

7. PLOS authors have the option to publish the peer review history of their article (what does this mean?). If published, this will include your full peer review and any attached files.

Reviewer #4: No

Reviewer #5: No

---

## [Editor Report · Acceptance letter]

1 Sep 2024

PONE-D-23-43492R2 

PLOS ONE

Dear Dr. Li, 

I'm pleased to inform you that your manuscript has been deemed suitable for publication in PLOS ONE. Congratulations! Your manuscript is now being handed over to our production team.

Kind regards, 

on behalf of

Dr. Pankaj Bhardwaj 

Academic Editor

PLOS ONE